



# The prediction of floods in Venice: methods, models and uncertainty

Georg Umgiesser[1,2], Marco Bajo[1], Christian Ferrarin[1], Andrea Cucco[3], Piero Lionello[4], Davide Zanchettin[5], Alvise Papa[6], Alessandro Tosoni[6], Maurizio Ferla[7], Elisa Coraci[7], Sara Morucci[7], Franco Crosato[7], Andrea Bonometto[7], Andrea Valentini[8], Mirko Orlic[9], Ivan D. Haigh[10], Jacob Woge Nielsen[11], Xavier Bertin[12], André Bustorff Fortunato[13], Begoña Pérez Gómez[14], Enrique Alvarez Fanjul[14], Denis Paradis[15], Didier Jourdan[16], Audrey Pasquet[16], Baptiste Mourre[17], Joaquín Tintoré[17], and Robert J. Nicholls[18]

[1] ISMAR-CNR, Institute of Marine Sciences, National Research Council, Venice, Italy
[2] Marine Research Institute, Klaipeda University, Klaipeda, Lithuania
[3] CNR—IAS, National Research Council, Institute for the study of Anthropic impacts and Sustainability in the marine environment, Oristano, Italy
[4] DiSTeBA - University of Salento and CMCC, Lecce, Italy
[5] Ca' Foscari, University of Venice, Venice, Italy
[6] CPSM, Centro Previsione e Segnalazione Maree - Protezione Civile, Venice, Italy
[7] ISPRA, Istituto Superiore per la Protezione e la Ricerca Ambientale, Venezia, Italy
[8] Arpae-SIMC – Agency for Prevention, Environment and Energy of Emilia-Romagna, Hydro-Meteo-Climate Service, Bologna, Italy
[9] Department of Geophysics, Faculty of Science, University of Zagreb, Croatia
[10] School of Ocean and Earth Science, National Oceanography Centre, University of Southampton, UK
[11] DMI – Danish Meteorological Institute, Copenhagen, Denmark
[12] UMR 7266 LIENSs, CNRS-La Rochelle University, 2 rue Olympe de Gouges, 17000 La Rochelle, France.
[13] Laboratório Nacional de Engenharia Civil, Lisbon, Portugal
[14] Puertos del Estado, Madrid, Spain
[15] Direction des Opérations pour la Prévision, Département Marine et Océanographie, Météo-France, Toulouse, France
[16] Shom (Service hydrographique et océanographique de la Marine), Toulouse, France.
[17] SOCIB, Balearic Islands Coastal Observing and Forecasting System, Mallorca, Spain
[18] Tyndall Centre for Climate Change Research, University of East Anglia. Norwich, UK

*Correspondence to*: Georg Umgiesser (georg.umgiesser@ismar.cnr.it)

**Abstract.** This paper reviews the state-of-the-art in storm surge forecasting and its particular application in the northern Adriatic Sea. The city of Venice relies crucially on a good flood forecasting system in order to protect the extensive cultural heritage, their population, and their economic activities. Storm surge forecasting systems are in place to warn the population of imminent flood threats. In the future, it will be of paramount importance to increase the reliability of these forecasting systems, especially with the new MOSE mobile barriers that will be completed by 2021, and will depend on accurate storm surge forecasting to control their operation. In this paper, the physics behind the flooding of Venice is discussed, and the state of the art of European storm surge forecasting is reviewed. The challenges that lie ahead for Venice and its forecasting systems are analyzed, especially in view of uncertainty. Some extreme events that happened in the past and were particularly difficult to forecast are also described.


## 1 Introduction

The city of Venice is situated inside the Venice Lagoon, which is connected to the Adriatic Sea by three inlets. Due to its low elevation with respect to mean sea level, for centuries the city was subject to occasional floods due to storm surge events, called Acqua Alta. During the last century the frequency of flooding has steadily increased (Ferla et al., 2007; Lionello et al., this issue), driven by relative sea-level rise which in part reflects climate change driven mean sea level rise, which in Venice is reinforced by local land subsidence (Carbognin et al., 2004; Zanchettin et al.,this issue).

The last three years have seen some of the most devastating high water events that Venice has ever experienced. In October 2018 a storm produced a sea level of 156 cm over datum. In November and December 2019, a series of storms created five high tides with water levels exceeding 140 cm. These events are called exceptional because only 25 such events have been observed in the last 150 years. They flood nearly 60 percent of the pedestrian walkways in the city and cover S. Mark's square with 60 cm of water. One of these high tides was the second highest ever measured water level at Venice being 189 cm and

only exceeded by the terrible flood in 1966 at 194 cm (Cavaleri et al., 2020).

Venice is tackling the problem of high water flooding by building mobile barriers (called MOSE) at each inlet of the lagoon. These gates sit on the seabed and compressed air is introduced into them to empty them of water, and they rise up until they emerge above the water to stop the tide from entering the lagoon. Construction has been underway since 2004, and has been criticized by many due to the elevated costs and the long timescale of construction. However, it is the only intervention that

will be able to effectively defend Venice from flooding due to storm surges in the near and medium future (Consorzio Venezia Nuova, 1997). MOSE is presently being commissioned and is expected to be fully operational by the end of 2021.

The operation of the MOSE uses a complex protocol, including water level measurements, and atmospheric and oceanographic forecasting. In particular, the decision to close the MOSE will depend on the water level forecast, and the time of closure will be decided based on water level measurements inside the lagoon (Consorzio Venezia Nuova, 2005). Hence, water level

forecasting is one of the fundamental aspects of the closing procedure of the mobile gates. Improving the reliability of these forecasts will contribute to avoid missing or false closures due to too low or too high predictions. This will save money as missing closures could result in flooding and damage to the city of Venice (shops, low lying apartments), while false closures will unnecessarily delay ship passage for its industrial and touristic ports (Vergano et al., 2010). Moreover, the forecast is essential to warning the population from imminent danger to life and goods, and the preparation of boat travel and walkways

for pedestrians inside the city.

From a historical perspective, the need to alert the population, protect commercial activities and organize transports and pedestrian mobility during floods of the Venice city center was motivated by the development of surge prediction tools. Early studies were published in the 1970's, in the wake of the initiatives motivated by the response to the dramatic flood in November 1966 (Finizio et al., 1972; Robinson et al., 1973; Tomasin 1973; Tomasin and Frassetto, 1979). Development resumed in the





1990's, partially in association with the perspective of developing a system with the accuracy required for operating the MOSE system (Vieira et al., 1993; Lionello 1995; Petaccia and Sallusti, 1995; Lionello et al., 1998; Bargagli et al., 2002).

This paper provides a state of the art review of water level forecasting for the city of Venice. It presents the geographical setting of the city and the processes giving rise to the occurrence of high waters and flooding in Venice. It then reviews the forecasting capabilities across Europe where various institutions conduct operational forecasting, including for the Adriatic

Sea and the Venice Lagoon. Different methodologies are presented that, in addition to the ones already operational, could significantly improve the forecasting system for Venice, including the operation of the MOSE barriers.

## 2 Geographic setting

### 2.1 The Adriatic basin

The Adriatic Sea is an epicontinental basin within the Eastern Mediterranean Sea extending southeastward from (12°E, 46°N)

to (19°E, 40°N), approximately 780 km long and 120–200 km wide (see Fig. 3 in Lionello et al., this issue).The bathymetry is characterized by a shallow northern shelf with water depths lower than 80 m, gently sloping in depth towards the middle part up to 250 m in the Jabuka Pit. The Palagruza Sill, 170 m deep, marks the beginning of the Southern Adriatic, which extends up to Otranto Strait separating the basin from the rest of the Eastern Mediterranean Sea. In the Southern Adriatic the bathymetry reaches 1200 m at the bottom of a wide depression known as the Southern Adriatic Pit.

The general circulation of the basin is mainly cyclonic and primarily thermohaline, driven by a combination of surface buoyancy fluxes, river discharges, and exchanges through the Strait of Otranto (**Poulain and Cushman-Roisin, 2002**). Along the eastern perimeter, the Eastern Adriatic Current flows northward transporting saline Levantine Intermediate Water (Artegiani et al., 1989), while on the western border, the Western Adriatic Current transports fresher surface water from Northern Adriatic toward the Otranto Strait (Bigniami et al., 1990a,b;). Along this side of the basin, a deeper southward flow

is also generated during the winter by the outflow of dense-water masses generated in the northern part of the basin (Malanotte-Rizzoli, 1977; Artegiani et al., 1989). Changes in the sea surface elevation induced by geostrophic balance are negligible.

The tidal regime has been interpreted as co-oscillations with the Mediterranean Sea, forced through the Strait of Otranto, and amplified by resonance phenomena along its longitudinal direction from south to north (Polli, 1961). The tidal form factor, consisting of the ratio of the amplitudes of the major diurnal and semi-diurnal tidal constituents, reveals that the semidiurnal

tidal regime ($F<1$) prevails in the Adriatic Sea, although the tide maintains a diurnal character ($F>3$) in the proximity of the semi-diurnal amphidromic points located in the middle of central Adriatic Sea (Ursella and Gacic, 2001, Lovato et al., 2010, Ferrarin et al., 2017).

Over the Adriatic Sea, between October and March, meteorological disturbances are frequently and usually characterized by the passage of low pressure systems or cyclones. These are often associated with strong southeasterly winds, the Sirocco, as



the cyclone approaches from the west, and with northeasterly katabatic winds, the Bora, as the cyclone travels eastwards. These two winds represent the dominant winds regimes during the winter period. In particular, the Sirocco winds blow along the main axis of the basin, drag the water towards the northwest and raise the sea water levels in the northern part of the basin. This can trigger resonance phenomena of water levels known as seiches, which dominant oscillation mode has the period around 21 h and a decay time of around 3 days (Cerovecki, 1997, Raicich 1999).

## 2.2 The Venice Lagoon

The Venice lagoon is situated at the north-western end of the Adriatic Sea (see Fig. 3 in Lionello et al., this issue). It is connected to the sea by three inlets with a width ranging from 500 to 1000 m. The depth of these inlets range from 7 to 13 m, depending on the inlet. The lagoon, during spring tide, has peak discharge of around 20,000 $m^3$/s with the Adriatic Sea, a very high number, considering that the average of the Po discharge is only around 1,500 $m^3$/s (Gacic et al., 2004).

The lagoon itself is quite shallow, on average around 1.5 m. Most areas are only 80 cm deep, but there are deep tidal channels that cut through the very shallow parts. Another particularity of the lagoon are the salt marshes that cover around 15 % of the total area (Umgiesser et al., 2004).

The tidal wave propagates mainly along the deep channels, and then expands into the more shallow areas (Rinaldo, 2001). The delay between the signal at the beginning of the inlets and the city center is 40 minutes. What concerns the amplitude of the oscillation there is neither water level damping nor amplification, so the water level reaches the center of the lagoon nearly undisturbed (Ferrarin et al., 2015).

During the last 500 years major engineering works have been carried out inside the lagoon, including diversion of rivers, building of the inlet jetties and dredging both existing and new artificial channels (Gambolati and Teatini, 2014). One of these channels, the Petroli channel, which runs from the central inlet across the lagoon to the industrial area, has been indicated to have worsened the phenomena of Acqua Alta inside the lagoon. This is however not the case, because modeling studies show that the water level enhancement is limited to a few millimeters (Umgiesser, 1999). Much more influence (some centimeters of reduction) has resulted from the creation of reclaimed areas where fish farming is carried out inside the lagoon.

Apart from the local wind effect that can shift water masses inside the lagoon, and can create water level differences of up to 50 cm between the north and south end, the level at the city of Venice is mostly determined by the water level just outside the inlets (Zecchetto et al., 1997). However, this means that the water level forecast can be divided into two different stages: a first one where the water level is determined at the lagoon inlets, and a second one where the effects inside the lagoon are being taken into account.



### 2.3 The city of Venice

The city of Venice is located in the central part of the lagoon and is situated on a group of 118 small islands that are separated
by almost 160 canals (known as 'rii') having width from a few to tens of meters, and depth of 1-5 m. The average elevation of
these islands is less than 1 m above the mean sea level. The Grand Canal is the main watercourse of this intricate network,
shaping out a large "S". The overall length of the canal system is about 40 km and its surface corresponds to, approximately,
10% of the total urban area (Zonta et al., 2005). The hydrodynamic regime in the canals is driven by tidal forcing as a
consequence of phase lags and level gradients occurring at the city boundary. During the flood tide the flow is predominantly
from SE to NW, while the direction is reversed in the ebb (Coraci et al., 2007). Current speed in the network is low, with
average maximum values up to 25 cm/s throughout the tidal cycle.

In the last century the city of Venice faced an increase in frequency and intensity of flooding events that periodically submerged
parts of the old city center, due to the combined effect of climate change and subsidence (Baldin and Crosato, 2017; Carbognin
et al., 2004). The increasing frequency of inundations represent a serious problem for citizens, businesses and tourist activities.
In October 2018 and November 2019, Venice and the North Adriatic have been exposed to extreme marine conditions (sea
level, waves) in turn induced by extreme weather conditions (Morucci et al., 2020). A complete list of other extreme events
can be found in Lionello et al. (this issue).

These flooding events pose a threat not only to the artistic, cultural and environmental heritage, but also to the economic assets.
To be able to face these events and to manage their occurrence, it has been invested in the safeguarding of Venice through the
planning and building of flood barriers, many other structural measures as well as through forecasting operational systems,
measurement networks and extreme events research activities (Barbano et al., 2012; Demarte et al., 2007).

### 3. The physics behind flooding in Venice

This section provides an overview of processes that influence and dominate the water level in front of and inside the lagoon
of Venice. The processes considered are tides, wind, atmospheric pressure and seiches.

### 3.1 Tides

Fluctuations of Adriatic Sea level and currents at tidal frequencies are among the largest in the entire Mediterranean Sea. The
tidal regime in the Adriatic Sea has been interpreted as co-oscillations with the Mediterranean Sea, forced through the Strait
of Otranto, and amplified by resonance phenomena along its longitudinal direction from south to north (Cushman-Roisin et
al., 2001; Vilibić et al., 2017). Consequently, tidal dynamics are particularly evident in the northern Adriatic Sea, where the
tidal range reaches values of more than 1 m at spring tide. Only seven tidal constituents, four semidiurnal (M2, S2, N2 and
K2) and three diurnal (K1, O1 and P1), give a significant contribution to the evolution of sea surface elevation in the North
Adriatic, reaching a range of 1 m in the Trieste bay (Polli, 1961; Mosetti and Manca 1972; Cushmain et al., 2001, Orlic,





2001, Book et al., 2009), and resulting in a mixed, mainly semidiurnal tidal regime (Ferrarin et al., 2015 and references therein).

Tidal propagation in the shallow Venice Lagoon is controlled by many factors which define the relationship among the celerity of the tidal wave, the inertia of the water mass and dissipative forces due to friction. While propagating from the inlets to the lagoon, the tidal wave is deformed according to a relationship between local flow resistance and inertia and the characteristics of the incoming tidal wave. At present, the incoming tidal wave is amplified in the lagoon indicating that the driving and reflecting tides are close to the condition of resonance. As a consequence of natural and anthropogenic morphological changes

that occurred in the lagoon in the last century, the amplitude of major tidal constituents grew, with a consequent increase in extreme high sea levels in Venice (Ferrarin et al., 2015).

### 3.2 Wind action

Due to the shape and the surrounding orography, the Adriatic Sea, and in particular the northern Adriatic, is mainly characterized by two wind regimes: the southeasterly "Sirocco" and the northwesterly "Bora" (Pasarić et al., 2009).

The Sirocco is a warm and wet wind mainly associated with a minimum mean sea level pressure located in the north-western Mediterranean or in Tunisia-Sicily Channel. It is channeled along the main axis of the Adriatic Sea by the bordering orography and blows often along the whole length of the basin, usually with wind speed between 10 and 15 m/s (Lionello et al., 2012). For these reasons it causes high waves and rain and when a strong Sirocco blows over the Adriatic the water is piled-up in the northern part of the basin. In combination with tides it is the main driver of flooding in Venice and in general in the shallow

coastal regions. The Sirocco is generally considered a basin response with large spatial scales (Signell et al., 2005).

As the Sirocco is stronger and more frequent in the southern Adriatic, the Bora blows more often in the northern Adriatic. With its northeasterly flow, it is a katabatic wind (Grisogono and Belušić, 2009) that, due to the complex orography of the Dinaric Alps on the eastern Adriatic coast, creates fine-structured jets and lee wakes with strong sub-basin scale spatial gradients across the northern Adriatic, reaching gusts of 100 km/h and higher (till 200 km/h). Its strength is due to the gradient

created by warm air over the sea surface and the cold area over the coastal mountain.

Two types of Bora are known in the northern Adriatic: the "Bora scura" that is characterized by windy and rainy conditions caused by a cyclone centered over the south of Italy and a anticyclone over northern Europe; and "Bora chiara" that is a cold, violent, dry, gusty wind, and is typically associated to a strong anticyclone over central and eastern Europe, which conveys cold air falling on the Adriatic Sea surface through the various gaps and valleys of the Julian and Dinaric Alps.Both types of

wind events can generate large waves and storm surges in the northern Adriatic, and especially in Venice, with different coastal impacts depending on the different morphology that characterizes the eastern Italian coast and the western Croatian one (Ferrarin et al., 2020a)



### 3.3 Low pressure systems

The passage of cyclones over the Mediterranean Sea produces positive and negative sea anomalies, which reach their largest
amplitude in the north Adriatic Sea and in the Gulf of Gabes (Lionello et al., 2019). The synoptic conditions leading to floods
along the Adriatic coast (e.g. beside Venice, in Trieste and Dubrovnik) consist of a driving low pressure center over central
Europe and frequently of an associated cyclogenesis south of the Alps (Robinson et al., 1973; Trigo and Davis, 2002; Lionello
2005, Lionello et al., 2019). In fact, a cyclone generated in the north-western Mediterranean produces the largest storm surge
events, by a combination of inverse barometer effect and wind stress acting over shallow water (Lionello et al., 2012). The
characteristics of the wind field and the cyclone track determine which location in the Mediterranean is most affected (Lionello
et al., 2012, Meðugorac et al., 2018). The cyclogenesis in the western Mediterranean Sea is triggered by the interaction between
the frontal layer of the primary cyclone and the Alps and it evolves as a baroclinic instability perturbed by the orography (Buzzi
and Tibaldi, 1978, Buzzi and Speranza 1986, Speranza et al., 1985), which is further modified by the moist Mediterranean
environment (Benzi et al., 1997, Krichak and Alpert, 1998). The presence of an area with frequent cyclogenesis in the north
western Mediterranean Sea is evident in the climatological maps (e.g., Lionello et al., 2016).

### 3.4 Seiches

Seiches are standing waves that occur in enclosed water bodies – lakes, bays and channels. In the Adriatic Sea, distinction
could be made between basin-wide seiches (characterized by periods that are close to the inertial period and therefore
considerably influenced by the Coriolis force) and local seiches (having periods that are much smaller than the inertial period
and thus marginally influenced by the rotation of the Earth).

Investigation of Adriatic-wide seiches was pioneered by Kesslitz (1910), who inspected de-tided sea-level records in time
domain, by Vercelli (1941), who performed filtering of tide-gauge records, and by Bozzi Zadro and Porettti (1971), who
subjected sea-level time series through spectral analysis. These analyses, and their numerous sequels, showed that the
amplitudes of Adriatic-wide seiches could reach 50 cm and that the periods of two basic modes are close to 21.2 h and 10.9 h.
The periods were verified by applying numerical modeling to the Adriatic Sea, one-dimensional (Sterneck, 1919) and two-
dimensional (Accerboni and Manca, 1973) with nodal line imposed in the Otranto Strait, and two-dimensional (Schwab and
Rao, 1983) with the Adriatic being considered as a part of a wider Mediterranean Sea. Numerical modeling expanded over the
subsequent years and, moreover, the question of generation and decay of the Adriatic seiches received some attention. Thus,
Raicich et al. (1999) confirmed the sequence of events usually leading to the generation of basin-wide seiches: during the first
phase, a storm surge develops in the Adriatic when a cyclone approaches the basin and thus exposes it to a combined action
of low air pressure and Sirocco wind; during the second phase, seiches are triggered in the Adriatic when the cyclone leaves
the basin and therefore air pressure rises while Sirocco wind slackens or veers to Bora. It was also pointed out that the partition
of energy between various normal modes depends on the spatial variability of Sirocco wind and the Adriatic bathymetry.
Cerovečki et al. (1997) considered twelve pronounced episodes of basin-wide seiches and obtained 3.2 ± 0.5 days for the free





decay time. It was concluded that the longevity of the Adriatic seiches could be ascribed to the weak influence of bottom friction and the small energy loss to the Mediterranean Sea. Due to their considerable amplitudes and persistence, the Adriatic-wide seiches could influence the flooding of the north Adriatic coastal area: if a cyclone approaches the Adriatic while the seiches related to the previous cyclone still last, a constructive or destructive superposition of the storm surge and the preexisting seiches may occur.

Local seiches were also found to be important in the Adriatic Sea. More specifically, the east Adriatic archipelago, with its numerous bays and channels, represents an ideal setting for development of short-period seiches. An overview of these waves, and of their importance in the generation of meteotsunamis, is provided by Vilibić et al. (2017). Much less is known on local seiches along the west Adriatic coast. Thus, for example, seiches may be expected to occur in the Venice Lagoon, with a period of 2–3 h defined by the length and depth of the basin. It appears, however, that the existence of such seiches and their possible

influence on the flooding of Venice did not receive much attention. In a rare paper addressing the subject, the lack of interest was attributed to the small depth of the lagoon (the mean value being close to 1 m) and the consequent strong damping of local seiches (Zecchetto et al., 1997).

Mean sea level rise will affect Adriatic seiches and tides depending on the adopted strategy of coastal defense (Lionello et al., 2005; Haigh et al., 2020). If dams are built to preserve the current coastline, increased wave speed and reduced bottom friction

will shorten the period of seiches and move it away from those of tides, whose amplitude would be consequently reduced. On the contrary, if the Adriatic Sea will be allowed to freely expand over coastal areas, the period of seiches will become closer to those of tides, whose amplitude would be, consequently, increased. Changes of amplitude will be of order of 10% for 1 m SLR, increasing substantially for larger values.

**4. Storm surge modeling in Europe**

In this section we describe existing storm surge modeling systems for Europe. It gives an overview of the capabilities and the peculiarities of the various areas and their forecast systems. Finally, the forecasting systems for the Adriatic Sea and the Venice lagoon will be discussed.

**4.1 The Atlantic Coast**

Along the European Atlantic coast, storm surges usually increase northward (Vousdoukas et al., 2016, Fortunato et al., 2016a).

This behavior is due firstly to the higher intensity of storms, although tropical hurricanes may exceptionally reach the Iberian Peninsula and drive large storm surges as well (e.g., Fortunato et al., 2017). Secondly, the width of the continental shelf increases northward, from a few tens of kilometers along the Iberian coast to a few hundred to the West of the English Channel and the British Isles. Coastal zones bordered by large shelves have extensive shallow areas, which causes the wind-driven surge to be much larger than in coastal zones bordered by deeper waters. Devastating storm surges are therefore more common



in the north. For instance, coastal flooding with inundation levels above 0.5 m occurred about once a year in Ireland between 1961 and 2006 (O'Brien et al., 2018). Although less frequent, catastrophic events have also occurred in the French (Breilh et al., 2014) and Iberian (Freitas and Dias, 2013) coasts.

These events have fueled the development of hindcast (Sebastiao et al., 2008; Bertin et al., 2014; Fortunato et al., 2017) and forecast (Fortunato et al., 2016b, Sotillo et al., 2019, Oliveira et al., 2020) models of tides and surges in the European Atlantic

coasts. Official forecasts are provided by several institutions, although the details of the models' implementations are not always readily available. In Portugal, the Laboratório Nacional de Engenharia Civil produces 48 hour forecasts of sea levels due to tides and surges using SCHISM (Zhang et al., 2016) in depth-averaged mode. The application is forced by FES2012 and GFS, and the grid has a resolution of about 250 m along the Portuguese coast (Fortunato et al., 2016). In Spain, Puertos del Estado runs several forecasts, described in section 4.4, while Meteogalicia runs ROMS at regional scales, with nested

applications of MOHID (Mateus et al., 2012) at estuarine scales. In France, Météo-France uses a 2DH barotropic application of the HYCOM model (Bleck, 2002), with a curvilinear grid of resolution down to 500 m along the coast. Compared to the Mediterranean model described in section 4.4, the Atlantic model includes tides and uses a spatially varying bottom friction. The British Met Office uses the NEMO model (Madec and the NEMO team, 2016), with a spatial resolution of 7 km and a Charnock approach with a tuned Charnock parameter to compute surface stress. A refinement to 1.5 km and a wave-dependent

approach to compute the surface stress are currently under development.

Simulations of tides and surges decoupled from waves are still the norm (Fortunato et al., 2016b, Fernandez-Montblanc et al., 2019). In particular, the agencies responsible for the production of official forecasts appear not to use coupled circulation-wave models in their routine predictions. However, the importance of waves on the sea levels has been widely demonstrated. Bertin et al. (2012, 2015) showed that the increased roughness of the sea surface associated with steep young waves can

significantly increase the wind stress, thereby enhancing wind-induced setup. The importance of wave-induced setup on the overall surge along the coast is also significant (Dodet et al., 2019, Idier et al., 2019). Hence, the interactions between waves and circulation will likely be increasingly included in forecast services. Also, most flooding studies continue to treat separately the hazards associated with high river flows and storm surges. However, the compound hazards should be considered (Ganguli and Merz, 2019, Khanal et al., 2019), and recent efforts are targeting the simulation of flooding induced simultaneously by

high river discharges and storm surges (Ye et al., 2020).

### 4.2 The North Sea

The North Sea has a long history of coastal flooding (Haigh et al., 2017). From historic accounts it has been suggested that large numbers of people were drowned around the coastlines of the North Sea in the years 1099, 1421, 1446, 1530, 1570 and 1717. It has been suggested that in the order of 10,000 people lost their lives in each of these events, but there is large

uncertainty in these numbers. In the last century, more than 2,000 people were killed along the coast of east England, the



Netherlands and Belgium during the 'Big Flood' of 31 January–1 February 1953 and 315 people were killed in Hamburg, Germany during the flood of 16-17 February 1962. These two events provided a major incentive for establishing storm surge forecasting and warning services for North Sea countries and were the driving force for major improvements in sea-defenses, e.g., the Thames Storm Surge Barrier in London, UK (Gilbert and Horner, 1986) and the Delta Works in the Netherlands
(Gerritsen, 2005).

Flather (2000) provided a review (now somewhat outdated) of the operational systems used for real-time prediction of storm tides and waves in northwest Europe. The Norwegian Meteorological Institute (DNMI) uses a 4 km resolution barotropic ROMS (Regional Ocean Modelling System) model, forced with ensemble weather predictions from the European Centre for Medium-Range Weather Forecasts (ECMWF) to forecast storm tides (Saetra et al., 2018). The Danish Meteorological Institute (DMI)
runs two storm surge models covering the North Sea and Baltic Sea, to provide storm tide forecasts for Danish Waters (Flather, 2000). In Germany, the Bundesamt für Seeschifffahrt und Hydrographie (BSH) uses nested storm surge model, forced with meteorological predictions from global and local area models of the Deutscher Wetterdienst (DWD), to provide forecasts of water levels (Dick, 1997). Verlaan et al. (2005) provide an overview of developments in operational storm surge forecasting in the Netherlands, where accurate forecasting of water level is very vitally important because large areas of the land lie below
mean sea level. Water level forecasts are made by the Dutch storm surge warning service (SVSD) in close cooperation with the Royal Netherlands Meteorological Institute (KNMI). These are based on the Dutch continental shelf model (DCSM) forced with forecasts from the meteorological high-resolution limited area model (HiRLAM). Since the early 1990s, a Kalman filter has been used to improve the accuracy of the forecasts by incorporating observations from tide gauges. In the UK, a Storm Tide Forecasting Service (STFS), which in 2009 became known as the UK Coastal Monitoring and Forecasting (UKCMF)
Service, was established as a direct result of the 1953 floods (Flather, 2000). Presently, the operational system uses the CS3X hydrodynamic model (Flather, 1994), which has a resolution of ~12 km, forced with the Met Office's Global and Regional Ensemble Prediction System (MOGREPS). In the future the CS3X model will be replaced with a NEMO-surge tide model (O'Neil et al., 2016). Recently, Fernández-Montblanc et al. (2019) describe efforts to create a pan-European Storm Surge Forecasting System (EU-SSF). This uses the SCHISM (Semi-implicit Cross-scale Hydroscience Integrated System Model)
unstructured grid model, forced with atmospheric fields from ERA-Interim as well as by an ECMWF high resolution forecast. Although not yet part of any operational system, Artificial Neural Networks (ANNs) have been tested for providing short-term forecasts of extreme water levels around North Sea coasts (e.g., French et al., 2017).

### 4.3 The Baltic Sea

The Baltic Sea is a brackish estuary, sometimes referred to as the world's largest fjord system. Connected to the world ocean
only via narrow straits, and further through the shallow North Sea, this large (1000 km scale), shallow semi-enclosed basin experiences storm surges mainly as a consequence of regional weather, i.e. wind and atmospheric pressure over the Baltic Sea itself (Samuelsson and Stigebrandt, 1996; The BACC II Author Team, 2015). The wind may shift a significant amount of


water from one end of the basin to the other, leading to a seesaw of sea surface elevation and depression. Surges may reach 2-2.5 m, but at least one extreme event in excess of 3 m has been reported and studied (Rosenhagen and Bork, 2009). While the

Baltic Sea is (almost) devoid of tides, standing waves (seiches) may occur. The so-called preconditioning, referring to the Baltic mean sea level (msl) during a storm surge event, plays a crucial role (The BACC II Author Team, 2015; Hupfer and others, 2003). Persistent North Sea westerlies may increase the Baltic msl by up to ¾ meter over a period of weeks. The transport capacity of the connecting straits is very limited, so during the relatively short course of a high wind event, the Baltic water volume may be regarded as almost constant. The size, and the peculiar shape, of the Baltic leads to the infrequent

occurrence of 'silent surges', whereby sea level may rise in parts of the basin under locally perfectly calm conditions, with high winds being limited to far away reaches. The U-shape of the western Baltic in particular leads to surges being built up from two sides (the central Baltic Sea and the North Sea) simultaneously.

These geographical features affect the specifics for Baltic hydrodynamic storm surge models. One may choose to model as a closed or as a semi-enclosed basin. The closed-basin (or parameterized boundary) approach is sometimes used for theoretical

studies. The operational model efforts in the Baltic countries regard the Baltic Sea as semi-enclosed, including the North Sea either in its entirety or partly. Several of the Baltic countries (Germany, Sweden and Denmark) have extra-Baltic coastlines as well, and benefit from having one connected-seas system. This kind of set-up requires high resolution of the connecting straits of down to 1 km width. A method to accomplish this while avoiding excessive computational load is local grid refinement, either by two-way nesting or by unstructured grids.

The inclusion of full thermodynamics and high vertical resolution improves sea level and surge simulation. The water exchange through connecting straits is often bi-directional, with opposing flow at surface and seabed, and thus not well described by a single-layer model. In winter, a large part of the Baltic is ice-covered. This modifies the uptake of energy from the wind. In order to establish a realistic estuarine circulation, fresh-water discharge of a catchment area three times the size of the Baltic is often included in some detail, depending on the wider range of model application.

High-resolution weather forcing is required for an accurate sea level forecast. Limited area models usually have a forecast range of just a few days, and constitute the basis for storm surge warnings in the region. Sea level predictions based on global weather forcing, extending beyond that time range, may be used for pre-warning. An alert may be raised, but no practical action is being taken.

Several model codes are used to model Baltic Sea storm surges operationally; among these the Baltic community model HBM

(Berg and Poulsen, 2012) and the European NEMO model (Hordoir et al., 2019; Madec and NEMO System Team, 2016). These two models constitute the model complex in Copernicus Marine Service for the Baltic Sea. The operational forecast models are used for national warning systems for storm surges, generally with an accuracy of 10-20 cm for 1-20 year events



### 4.4 The Western Mediterranean

Two decades ago there was little concern about storm surges in the Western Mediterranean Sea, a micro-tidal region with a meteorological signal magnitude similar or slightly larger than the tide signal. However, climate change impact on mean sea level rise and possibly on the intensity of the storms has yielded increased coastal erosion, damage to infrastructures and flooding events. Meteotsunamis are also a threat in this area, especially in the Balearic Islands. Therefore, forecasting total sea level, where traditional tidal tables are not sufficient, is now perceived as a need and even a critical tool for harbor operations and navigation.

Below a brief description of existing operational forecasts of sea level and storm surge in the region is presented.

*Puertos del Estado operational storm surge system (Nivmar):* The Nivmar system has been running since 1998 (Álvarez-Fanjul et al., 2001), based on a 2D-barotropic implementation of the HAMSOM model (Backhaus and Hainbucher, 1987). It is forced today with six-hour atmospheric pressure and wind fields extracted from ECMWF 1/8° hourly meteorological forecasts. Total sea level forecast at each harbor is provided by adding the tidal signal derived from a tide gauge station. A nudging scheme makes use of near-real time tide gauge data to correct low frequency signals not present in the barotropic model. In the framework of the ECOOP European project, a multi-model approach to provide probabilistic forecasts was first tested in 2008 (Pérez-Gómez et al., 2012). This system combined the output of existing storm surge and circulation (baroclinic) models at that time already operating in the region. This approach has been upgraded and became operational in the framework of the national project SOPRANO, combining the new CMEMS operational models: IBI-MFC (Sotillo et al., 2015) and MED-MFC (Clementi et al., 2019) with the Nivmar solution (Pérez-González et al., 2017, Pérez-Gómez et al., 2019). Every day, early in the morning, a deterministic forecast is provided by the old Nivmar solution. In the afternoon, the CMEMS forecasts are integrated with the tide gauge data and, by means of the Bayesian Model Average (BMA) technique, a probabilistic forecast band is generated for each harbor (Fig. 1).

*Météo-France operational storm surge system:* The Météo-France storm surge model has been in operation since January 14th, 2014. The model, developed by SHOM and Météo-France in the framework of the French HOMONIM project (Historique, Observation, MOdélisation des Niveaux Marins), is based on a barotropic version of the HYCOM code (HYCOM-2D: https://hycom.org/) (Pasquet et al., 2014, 2017). In the Western Mediterranean the maximum resolution of the model is 1.5 km, tides are not included, and the friction coefficient is constant in the domain. The system is launched 15 times per day with different atmospheric forcings, to allow the forecasters to assess the impact of each forcing on the sea level (Fig. 2). The model was validated with 22 storm events and one year of data, to calibrate the wind drag coefficient (wind stress according to the Charnock formulation) (Casitas et al., 2018). The one-year simulation presents a mean negative bias of -2 cm; during storms



this negative bias reaches -10 cm, with a mean time error of 34 min. Since 2016 an ensemble prediction system is available,
with 35 members of the Arpege Ensemble Forecasting System as forcings, run at 6:00 and 18:00 UTC, with forecast horizons
of 90h and 108h respectively.

*High-frequency sea level predictions associated with meteotsunamis in the Balearic Islands:* A high-resolution atmosphere-ocean modelling system is also implemented at SOCIB (Tintoré et al., 2013, 2019) to generate daily predictions of extreme
sea level oscillations associated with high-frequency atmospheric pressure variations (meteotsunamis) along the coasts of the
Balearic Islands. The modelling system (Renault et al., 2011; Ličer et al., 2017; Mourre et al., 2020) uses multiple nested grids
of the WRF and ROMS models for the atmospheric and oceanic components, respectively, with a 2-minute temporal resolution.
It specifically focuses on the harbor of Ciutadella in Menorca, which is the place most affected by these meteotsunami events
(also locally known as rissaga: Tintoré et al., 1988; Monserrat et al., 2006, Jansà et al., 2007), which occur several times per
year, mainly during the spring, summer and early fall seasons.

## 4.5 The Adriatic Sea

Although scientific works on marine circulation in the Adriatic Sea are very numerous and analyze it in detail, papers
presenting storm surge operational systems are few. Due to the importance of Venice, many systems were developed in several
operational Italian Institutions. Meteotsunamis are also an important concern in the Adriatic Sea, which has motivated the
recent development of dedicated ocean-atmosphere modelling systems (Denamiel et al., 2019).

The sea level in Venice has been recorded for more than 100 years and in this period there have been various extreme events,
such as in 1966 (Cavaleri, 2010) and 1979 (Grazzini, 2006), but almost every year events of a certain intensity can happen
(https://www.comune.venezia.it/node/6145), like in the last two years 2018 and 2019 (Cavaleri et al., 2019, 2020). The
following list contains the storm surge forecasting systems in the Adriatic Sea, divided according to the Institution in which
they operate:

**The Institute of Marine Sciences (ISMAR)** of Venice (Italy) is part of the National Research Council (CNR) and has been
developing a hydrodynamic model called Shallow water HYdrodynamic Finite Element Model (SHYFEM) for over thirty
years. SHYFEM is an open source finite element model available on Github (https://github.com/SHYFEM-model/shyfem).
This model has been used many times for storm surge prediction and many of the following operational systems are based on
it.

*Kassandra*: This system forecasts the total sea level and waves in the Mediterranean and the Black seas (Ferrarin et al., 2013).
The system is based on the SHYFEM model, coupled to the Wind Wave Model II (WWMII), and the forecast is available
online (www.ismar.cnr.it/kassandra). The resolution is variable but remains around 5 km throughout the Mediterranean and 1



km along the Italian coast. The model considers both the astronomical contribution to the sea level, with the calculation of the gravitational potential, and the wave set-up, thanks to the two-way coupling with the wave model;

*Tiresias*: This system, based on the SHYFEM model, uses a baroclinic formulation of the shallow water equations, and is therefore able to predict the temperature and salinity. The system extends to the whole Adriatic Sea and includes, in a single grid, the main lagoons (including the Venice Lagoon) and the Po delta (Ferrarin et al., 2019). This peculiarity allows the reproduction of effects such as the saline wedge intrusion in the Po delta, and the water circulation in three dimensions in the Adriatic;

*ISSOS*: (Ismar Storm Surge Operational System). Compared to the two previous systems, ISSOS, also based on the SHYFEM model, is focused on storm surge prediction. It operates with a computational grid, extended to the Mediterranean, with a lower resolution in order to reduce calculation times. Furthermore, the astronomical tide is not simulated but added locally where a total sea level forecast is needed. Although the simulation is barotropic and in two dimensions, the accuracy on the sea level is not affected. A second two-dimensional barotropic simulation is carried out in cascade to the first one to extend the forecast

into the Venice Lagoon, using the forecast of the Mediterranean simulation with tide (Ferrarin et al., 2020a). Finally, a slower three-dimensional baroclinic simulation is performed in the lagoon, using the same grid of the second simulation, in order to predict the temperature and salinity too. In this last simulation the temperature and salinity boundary conditions are retrieved by the Copernicus Marine Service.

**The Centro Previsioni e Segnalazioni Maree (CPSM)** is in charge of sea-level forecasts and warning, and is part of the
municipality of Venice (Italy). This center has the task of issuing the official forecast, with updates sometimes hourly, and alerting the population using sirens scattered throughout the inhabited centers in the lagoon. An example of the daily forecast page is given in Fig. 3. A statistics of the operational forecast in CPSM has also been published (Zampato et al., 2016).

*SHYMED*: (SHYFEM on MEDiterranean). This system, based on the SHYFEM model, updates previous systems that are no longer operational (Bajo et al., 2007, Bajo and Umgiesser, 2010). Its structure is similar to ISSOS (without the baroclinic
simulation) but uses different atmospheric forcing. In addition, there are three versions with various wind stress formulations and different corrections of the forecast wind. The forecast is emitted every twelve hours, but every hour the forecast is updated by means of a one-dimensional Kalman filter, using the latest observations (Bajo, 2020);

*HYPSE*: is a two-dimensional finite difference model, running over the Adriatic Sea, with assimilation of one sea-level station (Lionello et al., 2006). This system is not operational anymore;

*System based on Delft-3D:* This was a system implemented some years ago but is not operational either.


**The Italian National Institute for Environmental Protection and Research (ISPRA)** is an institution under the umbrella of the Italian Ministry of Environment. The modeling system *SIMM* (Sistema Idro-Meteo-Mare) is a chain of operational systems that includes the atmosphere, waves and storm surge predictions (Speranza et al., 2007, Mariani et al., 2015). The storm surge forecasting system is based on the SHYFEM model, which is set to execute two-dimensional barotropic simulations. There are different versions that use different spatial resolutions, meteorological forcings and data assimilation (DA). The version with DA is based on the dual 4D-Var technique (Bajo et al., 2017) and assimilates sea-level data from nine tide gauges along the Italian Adriatic coast.

**ARPAE** is the Regional Agency for Prevention, Environment and Energy of Emilia-Romagna (Italy). The Hydro-Meteo-Climate Service of Arpae (Arpae-SIMC) actively works in numerical model forecasting, both deterministic and probabilistic, at different time scales from very short-range, in case of extreme events, to seasonal scale. Arpae-SIMC is aCentre of Competence for the Italian National Civil Protection system as well as the Support Centre for the Civil Protection Agency of Emilia-Romagna.

The integrated modelling system developed by Arpae-SIMC for its marine and coastal activities (Fig. 4) consists of a wave-forecasting operational chain called MEDITARE (Valentini et al., 2007) and AdriaROMS (Chiggiato and Oddo, 2008, Russo et al., 2013), which uses the ROMS model implemented in the Adriatic Sea (Shchepetkin and McWilliams, 2005). Both systems are forced by the non-hydrostatic limited-area COSMO-I meteorological model (WetterDienst (DWD); 2004).

**Copernicus Marine Environmental Service (CMEMS)** is the marine section of the European Copernicus program. This program provides remote sensing and modelling data, freely available on the web (https://marine.copernicus.eu/). The modelling product for the Mediterranean Sea is called Med-MFC Analysis and Forecasting Physical System (MedFS). *MedFS* is a coupled hydrodynamic-wave (NEMO-WW3) system (Clementi et al., 2017b,a) with a data assimilation component (Storto et al., 2015). The horizontal grid resolution is 1/24˚, while the vertical levels are 141. The model solves the three-dimensional baroclinic shallow water equations, without the tidal component. Only the surge component is published at the web site.

**Slovenian Environmental Agency (ARSO)** is part of the Slovenian Ministry of the Environment and Spatial Planning. *SMMO* is a forecasting system based on the Aladin atmospheric model and originally on the Princeton Ocean Model (POM), with a regular resolution of 1/30 of a degree. Its computational domain is limited to the Adriatic Sea (Licer et al., 2016). The system has been recently upgraded to the NEMO ocean model with a resolution of 1/216 of a degree (personal communication).

**Consorzio Venezia Nuova (CVN)** is the Italian consortium of Companies responsible for building the Venice flood barrier system, named MOSE. They are also running a forecasting system for the operation of the mobile barriers. The storm surge model at CVN is developed by DHI and is based on Mike 21 (Vieira et al., 1993). Data from two atmospheric models are used (ECMWF and COSMO 5M). The modeling chain consists of a Mediterranean model that drives an Adriatic Sea model and in





turn the Venice lagoon model. The forecasting period varies between 18 hours and 5 days (https://www.mosevenezia.eu/control-room/). The output of this forecasting system is not made public.

Many of the aforementioned models have been considered together in the I-STORMS multi-model system (Ferrarin et al., 2020a), which provides ensemble forecasts for water levels and waves in the Adriatic and Ionian seas.

**4.6 The Venice Lagoon**

Forecasting of storm surge events is mainly done in the Adriatic Sea as described in the last section. In order to resolve also the tidal and storm surge propagation inside the lagoon high resolution models have to be run inside the Venice lagoon. This can be done either in a traditional way, where the water levels computed outside the lagoon with models running on the Adriatic Sea are used to force a new model with a modeling domain resolving only the lagoon area, or a combined model with a domain

including both the Adriatic Sea and the Venice Lagoon. Both types of approaches are used in Venice, and are described below. Moreover, statistical models are also used for forecasting.

The CPSM is, for a long time (from 1981), running statistical models that use the pressure, wind, and water level data from past events to forecast the new events in the future. Especially for the short time range (up to 12 hours) this is quite effective. These models will be presented in the next section.

What concerns deterministic models that actually resolve the physics of the surge propagation, one of the first models that has been implemented at the CPSM was a forecasting system based on the finite element model SHYFEM (Bajo et al., 2007). The system, named SHYMED (see section 4.5), is implemented on the whole Mediterranean and uses atmospheric forcing from ECMWF and sea-level boundary conditions from Copernicus. A second simulation inside the lagoon uses the forecast of the first simulation and the tide recorded at the oceanographic tower "Acqua Alta", 8 nautical miles off the Venetian coast. In its

actual implementation the model does not use assimilation techniques. However, the model results inside the lagoon were corrected with a neural network approach in the past (Bajo and Umgiesser, 2010) and now with a unidimensional Kalman Filter (Bajo, 2020), which allows hourly updates of the forecast.

At ISPRA, another implementation of the SHYFEM model (SIMM) is running daily (Mariani et al., 2015). This application uses wind both from ECMWF and from another atmospheric model (BOLAM), produced by ISPRA in Rome. A version of

485 this implementation uses a dual 4D-Var assimilation technique where nine tide gauges in the Adriatic are used. Also in this case, a new model run propagates the water level inside the lagoon and publishes water levels for some important islands inside the lagoon, such as Venice, Burano and Chioggia.





At the CNR the ISSOS storm surge model is running. As the implementations of the models in CPSM and ISPRA, the model first resolves the Mediterranean Sea, and then computes the water levels in a second run inside the lagoon. A lagoon run of this model is baroclinic, therefore taking also into account temperature and salinity (Ferrarin et al., 2020a).

There is one model (Tiresias) that is implemented on a numerical grid that comprises the whole Adriatic Sea and the Venice Lagoon. With this model water levels are directly computed at the stations inside the lagoon, and there is no need to run a second model to propagate this information inside the lagoon (Ferrarin et al., 2019).

## 5. Challenges in forecasting flooding for the Venice lagoon

Here we stress the peculiarity of Venice for water level forecasting. We show what deterministic and statistical forecast models can do and how they complement themselves. We also discuss the importance of assimilation for improvement of the forecast and ensemble methods for the estimation of the uncertainty. We then also give some examples of past storm surge events.

### 5.1 Local effects

The Venice Lagoon has experienced significant subsidence effects (Baldin and Crosato, 2017; Carbognin et al., 2004) in the last century and is constantly exposed to the risk of flooding from storm surge, enhanced also through relative sea level rise (Frassetto, 2005). The semi-closed shape of the Adriatic Sea favors the occurrence of intense storm surge events and the sea level rises unusual values because of the local low-pressure system cyclogenesis (Genoa) and associated strong winds, driven by orography. It is well known that the hydrodynamics of the Venice Lagoon is strongly dependent upon both tides and prevalent local wind regimes (Berrelli et al., 2007).

The level at the Punta della Salute tide gauge (Venice center) is not enough to widely represent the conditions of the lagoon basin during occurrences of storm surge (Ferla, 2005). Sea-level observations have shown that the forcing action of the wind along the lagoon surface gives rise to considerable accumulations on the water against the southern or northern boundaries of the lagoon, depending on the wind direction. The maximum water levels are significantly different in Venice and in other larger inhabited centers, such as Chioggia in the southern section of the lagoon, or Burano in the northern portion. Under these varying weather conditions, sea level differences between the various parts of the lagoon and especially between lagoon and sea can determine asymmetrical hydrodynamic conditions at the inlets.

Local surges, mainly due to wind setup, have a scale of few kilometers inside the Venice Lagoon and can produce significant effects, especially near the lagoon borders; Bora wind (north-easterly) has relevant effects in the southern Lagoon, nearby Chioggia island, where fishing valleys and reclaimed areas are located, all situated below the mean sea level.

According to literature, an average difference of 50 cm has been estimated between the Northern and Southern part of the Lagoon (Fig. 5) in different cases of Bora wind of 5-7 m/s (Pirazzoli, 1981, Berrelli et al., 2006), and higher difference in case of extreme meteorological events (Cordella and Ferla, 2007). Such phenomena were observed mainly in February during the


years 2012, 2014 and 2015, when a strong Bora wind was blowing (ISPRA, 2012; Coraci and Crosato, 2014). On the other hand Sirocco winds (south-easterly), blowing towards the North Adriatic Sea along the main basin axis, push water masses

towards the Venice lagoon, particularly on the northern areas (Mariani et al., 2015).The combination of all these local effects and climate change (Rinaldo et al., 2008) could lead to severe and disastrous storm surge events, as happened during the last two years (October 2018, November 2019; Morucci et al., 2020; AAVV, 2020).

### 5.2 Deterministic forecast models

The deterministic operational models for predicting the sea level in Venice are typically storm surge models, which solve
shallow water equations in two or three dimensions. This simplified formulation is often used, instead of more complex equations, as it is extremely fast and has an accuracy comparable to more complex equations in storm surge resolution.

Many operational systems based on deterministic models use computational domains extended to the whole Mediterranean Sea, even if some of them run only the Adriatic Sea. However, the extension to the Mediterranean Sea avoids problems in the reproduction of the seiches, which have a nodal point near the Strait of Otranto (Cerovecki et al., 1997). The lateral boundary
conditions of sea-level and, sometimes, water flux can be retrieved by other operational models, such as those of the European Copernicus Programme (https://marine.copernicus.eu/). The surface boundary conditions are the most important for a storm surge model and consist of wind and pressure fields. Due to the orographic conformation around the Adriatic Sea, atmospheric models tend to have large errors and local-scale models with high resolutions may be required.

Deterministic models can include the tidal potential in their equations and thus simulate the propagation of the astronomical
tide. However, many operational systems prefer to simulate only the storm-surge component and locally add the astronomical tide, calculated through harmonic analysis. In this way the accuracy on the total level is not affected by the error in the tide modeling.

Others possible problems in the Adriatic Sea concern the complexity of the coasts and the need to adequately represent them (for this purpose it is better to use models with unstructured grids) as well as the lack of high resolution and accurate
bathymetric datasets in the continental shelf border. Finally, forecast scores can further be improved through the estimation of the forecast thanks to data assimilation and ensemble forecast techniques, based on ensembles of different forcing perturbations (Mel and Lionello., 2014b) or a multi-model approach (Ferrarin et al., 2020a).

### 5.3 Statistical forecast models

The statistical models represent a fundamental part of CPSM (Centro Previsione e Segnalazione Marea) tidal forecasting
modelling system. The model structure has been designed and calibrated using a database of tide and atmospheric data, over a period exceeding 50 years (from 1966 to 2016). The database contains the values of tide level, atmospheric pressure, baric gradients and wind in the Adriatic and Tyrrhenian Sea. These values were used to obtain regression coefficients for different versions of the models inspired by the autoregressive scheme, with expert type, from the ARMAX family. (Goldmann and Tomasin, 1971; Sguazzero et al., 1972; Tomasin, 1972)




The actual operational structure of the system of statistical models is illustrated in Table 1. The engineering processes of the expert structure divided the database into subsets of similar cases, considering: meteorological situation, seasonality, and moment of forecast. With this process, many classes of coefficients have been created. Every single class corresponds to an appropriate set of coefficients used by the basic model for statistical interpretation of the evolution of sea level in the Venetian area (lagoon and sea).

Another important development for the forecast of sea level with statistical models consists in using the expected pressure and wind fields. CPSM is using the forecast fields issued by ECMWF and COSMO-I. It allows us to make projections into the future, with an hourly step, using observations and predictions simultaneously. (Canestrelli and Pastore, 2000; Tosoni and Canestrelli, 2011)

Since 2009, a multi-model ensemble system has been operational which is able to collect different information making choices
in reference to historical performance of single statistical models running in CPSM. Nowadays, two multi-model-ensemble (MME) families are operating in CPSM, so designed:

1.    For every step of forecast, models are selected which have historical minimal errors in similar meteorological conditions and in the same season, and a weighted average in relation to Mean Absolute Error is calculated (Markowitz, 1952; Lintner, 1965).

2.    For each forecast step, a linear regression is constructed with the historical set of all forecasts for a specified class. The coefficients obtained are applied to the expected values (Black, 1972; Mossin, 1966; Sharpe, 1964)

### 5.4 Data assimilation in flood forecasting

In the specific case of the Adriatic Sea, data assimilation has already been applied simulating the operational practice and using the 4D-Var technique (Lionello et al., 2006). Lionello et al. (2006) have shown that the prediction of the north Adriatic storm
surge can be improved by adopting a suitable data assimilation procedure. They used the hourly sea level observations available at the "Acqua Alta" platform 8 nautical miles offshore the lagoon inlet. The assimilation procedure is based on the adjoint model (Lewis and Derber, 1985; Talagrand and Courtier, 1987; Thacker and Long, 1988; Thacker, 1988). The procedure has been shown to be capable of compensating for the forecast errors (including those caused by inaccuracy of meteorological forcing and model shortcomings).The reliability of the storm surge forecast has been consistently improved simulating one
month of operational prediction and reducing the error to 50% of the original value in the 1 to 3-day forecast range. However, the availability of only one sea-level station limited the capability of the method to correct any seiche oscillations extended along the entire Adriatic basin.

A similar technique, the dual 4D-Var, has been used more recently in an operational system (SIMM, sec. 4.5) to assimilate the residual level from nine stations along the Italian coast, in order to improve the forecast of the storm surge. Furthermore, the
same hydrodynamic model and data assimilation system were used to analyze the impact of the assimilation of altimeter data in two historical storm surge cases (Bajo et al., 2017). Although assimilation techniques based on 4D-Var are very advanced,


the major problem is that of prescribing a good background error covariance matrix which, instead, is automatically calculated using assimilation techniques based on an ensemble of simulation. A technique belonging to this group, the Ensemble Kalman Filter (EnKF), was used more recently in Bajo et al. (2019), showing a significant improvement in the reproduction of the

storm surge in two extreme events. In this case, the residual level from different coastal stations, both along the Italian and the Croatian coasts, was assimilated. The improvement was largely due to the presence of pre-existing seiches, which were better reproduced following the improvement of the initial state of the system.

The EnKF used in this paper and similar techniques, exploit the variance of an ensemble of simulations to construct the background error covariance matrix (Carrassi et al., 2018). If the ensemble is well perturbed the matrix is very realistic and,

moreover, it is variable over time. The ensemble can be calibrated so as to be "reliable", that is, its width is able to represent the error that the forecast will actually commit (further discussed in the next section). In this way, data assimilation is able to exploit the correction of the observations to improve the whole state of the model in a physically coherent way (Bajo et al., 2019).

Finally, at the CPSM center "local" data assimilation techniques are used, which are much simpler to apply. These techniques

are able to significantly improve the short-term forecast, originally made by a deterministic model, and to provide hourly updates of local forecast time-series, as in the case of a statistical model (Bajo, 2020).

### 5.5 Dealing with uncertainty – ensemble forecast

A dynamic model generates a deterministic and single storm surge forecast time series at each individual grid point. If the model is validated against in-situ tide gauges, the error or accuracy of the model at these specific locations can be assessed.

However, uncertainty of the forecast and its dependence on the forcing, model characteristics or bathymetry is usually unknown and possibly underestimated, with the consequences this may have for risk managers and decision-makers, who would preferably rely on probabilistic forecasts (forecast plus confidence interval).

The standard procedure for dealing with uncertainty has been used for years in meteorological forecasts (Leith, 1974; Hamill et al., 2000) and it is now strongly recommended in oceanography: the combination of different model solutions or ensemble

modelling. The ensemble prediction is a procedure implemented at ECMWF (Molteni et al., 1996) since 1992 in the operational prediction system to provide a probabilistic weather prediction. The high sensitivity of the predicted weather evolution (which can be characterized as a chaotic system) makes weather deterministically unpredictable beyond a finite time range (practically beyond 10 days). The deviation of wind and sea level pressure fields from their actual evolution will determine a corresponding deviation on the predicted sea level. Analogously the uncertainty of the predicted wind and sea level pressure fields will

determine an uncertainty in the evolution of the sea level. The Ensemble Prediction System (EPS) consists of a set of forecasts that are different because initiated from a set of different initial conditions, which are designed to represent the uncertainties in the knowledge of the state of the weather (Buizza and Palmer, 1995). If a set of different weather predictions are used to





drive different sea level simulations, the probability distribution function of the forecast sea level values provides a practical tool for estimating the uncertainty of the sea level forecast and the probability of exceeding a given sea level threshold.

Flowerdew et al. (2010, 2012) were the first to apply the EPS for operational surge prediction. In a different approach, multi-model storm surge ensemble prediction has been used for New York and the North Sea by Di Liberto et al. (2011) and Siek and Solomatine (2011), respectively. Ensemble forecasting capabilities have also been recently explored for the prediction of meteotsunamis in the Balearic Islands (Mourre et al., 2020), showing an interesting potential to quantify the uncertainties associated with the predictions.

Another option is the combination of different operational models, with different characteristics and even physics, but providing sea level forecasts at the same area (multi-model forecast). A multi-model storm surge forecast was first implemented in the North Sea, by combining the storm surge forecasts from different countries in the region in 2008. The system included the use of the Bayesian Model Average (BMA) statistical technique for validation of the different members and generation of a combined improved prediction, with a confidence interval (Beckers et al., 2008). This was the methodology

tested first by Puertos del Estado in 2008, for the Spanish coast (Pérez Gómez et al., 2012), combining in this case the output of an existing storm surge forecasting system (Nivmar, see section 4.4) and circulation (baroclinic) models at that time already operating in the region. Today this multi-model forecast is operational at Puertos del Estado and combines Nivmar with the CMEMS regional operational models: IBI-MFC (Sotillo et al., 2015) and MED-MFC (Clementi et al., 2019). A first deterministic forecast is provided by the old Nivmar solution early in the morning every day In the afternoon, the CMEMS

forecasts are integrated with the tide gauge data and, by means of the Bayesian Model Average (BMA) technique, a probabilistic forecast band is generated for each harbor (Pérez-González et al., 2017, Pérez-Gómez et al., 2019) (Fig. 1). A multi-model ensemble forecasting system has been recently developed for the Adriatic Sea combining 10 models predicting sea level height (either storm surge or total water level) and 9 predicting waves characteristics (Ferrarin et al., 2020a).

Storm surge ensemble prediction has been used to forecast sea level in Venice by Mel and Lionello (2014a). They used a 50

member ensemble to simulate 10 events showing that EPS slightly increases the accuracy of the prediction with respect to the deterministic forecast and that the probability distribution of maximum sea level produced by the EPS is acceptably realistic. They also showed that the storm surge peaks correspond to maxima of uncertainty and that the uncertainty of such maxima increases linearly with the forecast range. The same procedure was used for the simulation of the operational forecast practice for a three month long period (fall 2010) by Mel and Lionello (2014b).

Results have shown that uncertainty for short and long lead times of the forecast is mainly caused by the uncertainty of the initial condition and of the meteorological forcing, respectively. The probability forecast based on this ensemble technique has a clear skill in predicting the actual probability distribution of sea level. A computationally cheap alternative, called ensemble dressing method, has been proposed by Mel and Lionello (2016). It replaces the explicit computation of uncertainty by ensemble forecast with an empirical estimate. Instead of performing multiple forecasts, the procedure "dresses" the forecast





of sea level with an error distribution form, which includes a dependence of the uncertainty on surge level and lead time, on one hand, of the uncertainty of the meteorological forcing, on the other hand. It is shown that this computationally cheap alternative provides acceptably realistic results.

**5.6 Some examples of forecasting past storm surge events in Venice**

Due to the crucial effect of climate changes, in the last century the city of Venice faced an increase in frequency and intensity
of flooding events that periodically submerge parts of the old city center (Lionello et al., this issue). The highest water level occurred in 1966 (194 cm ZMPS) followed by two more recent events in October 2018 (156 cm ZMPS, Cavaleri et al., 2019) and in November 2019 (189 cm ZMPS, Cavaleri et al., 2020), the latter much worse, but the former potentially more dangerous (Morucci et al., 2020). According to Lionello et al. (this issue) the relative extreme sea level is composed by the contribution of various factors: astronomical tide, seiche, storm surge, long-lasting sea level anomalies due to planetary atmospheric waves
(PAW), meteotsunami, wind setup within the lagoon, inter-decadal, inter-annual and seasonal (IDAS) sea level variability, relative sea level rise. The combination of storm surge, meteotsunamis and PAW surge represents the direct action of the meteorological forcing on extreme sea levels and it is collectively termed surge.

In 1966, the astronomical tide at the nominal time of the peak was at its minimum (-10 cm) and the peak sea level was due to an exceptional direct meteorological contribution (143 cm, Table 1 in Lionello et al., this issue), mostly attributed to storm
surge induced by a strong and persisting southerly wind over the whole Adriatic Sea. It has also been extremely long in time: for 22 hours the level remained over 110 cm, and the surge was over 100 cm for more than 10 hours and over 50 cm for about 40 hours (De Zolt et al., 2006; Trincardi et al., 2016). The event of November 2019 represents the second higher storm surge, following the widespread flood event of 1966. De Zolt et al. (2006) and Roland et al. (2009) simulated the 1966 event with coupled wave-current models revealing the good accuracy of the model in reproduction of the sea level in the northern Adriatic
Sea and in Venice.

As for the 1966 case, the event of 29th October 2018 was due to a low pressure system in the western Mediterranean Sea and an intense southeastern Sirocco wind blowing for many hours along the Adriatic Sea. It showed a maximum surge contribution of 117 cm at 19:25 UTC (mostly due to storm surge and PAW surge), that happened during the minimum of astronomical tide. The development of the general meteorological pattern during the storm was well forecasted by the meteorological models
(Fig. 6) with a resulting good predictability of the peak sea level in Venice some days ahead (Cavaleri et al., 2019; Ferrarin et al., 2020a).

The 12th November 2019 exceptional event was due to the combined effect of many meteorological forcing: the unusual high level of the Mediterranean Sea in November, reflecting an anomalous general atmospheric depression over the basin; a deep low-pressure system over the central-southern Tyrrhenian Sea that generated strong Sirocco (south-easterly) winds along the
main axis of the Adriatic Sea; a small-scale atmospheric pressure minimum developed over the Northern Adriatic and moving





rapidly northward along the Italian coast; very strong winds over the Lagoon of Venice, which led to a rise in water levels and damages to the historic city (Cavaleri et al., 2020; Ferrarin et al., submitted). The peak of the meteorological contribution (100 cm) happened during the maximum of the astronomical tide with the disastrous effects of a total sea level of 189 cm. This event was strongly underestimated by all operational ocean forecasting systems. Such underestimation was mostly due the uncertainties related to the reproduction of the intensity and path of the small-scale cyclone travelling in the northern Adriatic Sea generating the meteotsunami along the coast and local setup within the lagoon. Ferrarin et al. (submitted) demonstrated that a relatively small error in the meteorological forecast (cyclone trajectory misplaced of about 10-20 km) may produce a relevant error in the sea level prediction in Venice which relies on accurate small-scale meteorological forcing.

The comparison between the mentioned events shows the important role played by the strength and timing of the different meteorological components of the sea level. The low-frequency meteorological components, i.e. inter-annual, seasonal and PAW oscillations, have minimal variations over a few days and can be easily considered into short-term sea level forecasts using near real time observations. The predictability of extreme high sea levels in Venice, therefore, resides on the model's capacity in reproducing storm surge, seiches and high-frequency oscillations (e.g., meteotsunami). As discussed above for the 1966 and 2018 events, the large-scale synoptic framework associated with important storm surge events is predictable several days in advance. Seiche oscillations can be correctly simulated in the Adriatic Sea, especially when applying data assimilation (Bajo et al., 2019). Instead, beta mesoscale phenomena are still not well reproduced by the commonly used low-resolution (>2 km) mesoscale models, and therefore their contribution to the sea level remains underestimated (Denamiel et al., 2019).

It is worth noting that the timing error (even of an hour) of the meteorological forecasts may have dramatic impacts on the expected overall sea level because of the combination of surge and tide. Ensemble forecasts could be a solution providing the statistical distribution of the combined possibilities (Cavaleri et al., 2019).

## 6. Discussion

Predictions of sea level variations have always been a challenging issue aimed at protecting the marine and coastal environment, especially in the North Adriatic Sea and in the Venice lagoon, that are increasingly exposed to the flooding risk from storm surges, well known as "Acqua Alta" phenomenon. Even though forecasting models may provide important information on the evolution of sea level due to storm surges, they are still imperfect and uncertainty in the future evolution of events plays an important role in decision making (Coccia, 2011). In order to better account for uncertainty, one of the viable possibilities is the use of ensemble forecasting. Using this technique a good indication of the uncertainty of a storm surge forecast is possible. Information of this kind is important to alert the population of risks concerning the flooding of the city. However, one thing is to have an indication of uncertainty, and another is the communication of this uncertainty to the population, which is another difficult task. The I-STORMS project has also elaborated on risk communication under uncertainty (Ferrarin et al., 2020a).



Data assimilation also offers an important tool to be exploited for simulations of the Venice lagoon. This technique has shown to be very successful, especially for a near future forecast (up to two days). However, even if in forecasts of a few hours the error can be kept quite low, this effect seems to vanish if results of longer forecasts are examined. It has been shown (Lionello et al., 2006) that the prediction of the north Adriatic storm surge can be improved by adopting a suitable data assimilation procedure. Other authors (Bajo et al., 2019) have shown that assimilation of data from many sea-level stations can improve the simulation of the timing of seiches in the Adriatic basin and therefore correct the initial condition of this phenomenon. A further potential improvement could be assimilation of altimeter data, especially if in the future their temporal frequency and their resolution near coastal zones will improve (Bajo et al., 2017; De Biasio et al., 2016; De Biasio et al., 2017).

Another possibility is the combination of deterministic forecasting (that allows for periods of days) with statistical methods that are stronger in the short range forecasting. The idea is to statistically correct the results from deterministic models with statistical methods such as neural networks (Bajo and Umgiesser, 2010) or local Kalman filters (Bajo, 2020). Results have shown improvements in the quality of the forecasts except for those events that are out of the statistics samples (through which statistical models are designed), and that are often wrongly interpreted by the method and show worse results.

Multi-model forecasts are another way to go forward. At CPSM a variety of models (deterministic and statistical) are running (Canestrelli and Pastore, 2000) and these model results can be used to provide a more reliable estimation of the future water level. This technique can be especially used to give more weight to models that are better estimators under special conditions, where other models are less reliable.

The concept of Predictive Probability is crucial and it is defined as the probability that an event will occur at a certain time into the future, conditional on prior observations and all the information available at the time of the forecast, and Krzysztofowicz, (1999) suggesting that prior information can be encapsulated into one or more model forecasts. In other words, the decision triggering threshold will not be based on different sea level thresholds (warning level, alert level, flooding level), but rather on different probabilities of a threshold to be overtopped (Coccia, 2011).

In this framework, the Model Conditional Processor (MCP-MT) has been implemented in its test configuration as hindcasting on Venice – Punta della Salute historical time series. It has been applied in a multi-model and multi-temporal form to the "Acqua Alta" forecast in the Venice Lagoon, both in order to assess the predictive probability of threshold exceedance within a given time horizon, as well as to the estimate the expected time of occurrence of a given future event.

A need for a good and reliable forecast is also requested by the consortium that operates the mobile barriers (MOSE) that have started to be in a pre-operational phase at the inlets. In October 2020 the barriers have already been closed three times and in these cases could protect the city from flooding. Complete operational functioning is estimated for the end of 2021. However, at this time with the data collected from the three closures, some conclusions can be drawn. As can be seen (Fig. 2 in Lionello et al., this issue), on the 15th of October 2020, a strong setup inside the lagoon happened, and the water level difference between the northern and southern lagoon exceeded 40 cm. A good forecast is needed for these situations in order to find the exact time for the closing of the barriers. Late closure might be able to avoid the high water at Venice, but the City of Chioggia at the southern end of the lagoon could experience flooding.


Once the MOSE is fully functional, a reliable forecast (not an exact one which is impossible at this stage) is needed in order to satisfy the various needs of the stakeholders. It is important to understand that the decision to close the lagoon is taken based on the forecast of the water level, wind and rain some hours before the event (Consorzio Venezia Nuova, 2005). Once the decision has been made to close, a water level is fixed that depends on the meteorological conditions (wind and rain), at which

the barriers will be closed. It is therefore the forecast that at the end will decide if the MOSE will be operated or not. And this makes the operational forecasting of paramount importance for the lagoon and the city of Venice (Umgiesser, 2020).

The port would avoid as much as possible closures due to a forecast which is too high with respect to the actual measured water levels. These false closures would create an interruption of the ship passages and the maritime traffic. On the other hand, the shop keepers would make absolutely sure that high water will not flood their shops and restaurants, trying to avoid as much

as possible missed closures because of a forecast which was too low. A 10 centimeter forecast error will not go unnoticed by a shopkeeper that expects a forecasted 105 cm of water level (where the MOSE will not be activated) and experiences an actual 115 cm of measured water level, especially if the shop floods at 110 cm. Another aspect are hotel owners that always fear a water level forecast that is too high because cancelling of reservations could result from these alarms.

It is clear that for many aspects reliable water level forecasting is needed in Venice. Operational systems are in place all over

Europe that show the state of the art that Venice should strive to follow in order to protect its inhabitants, warn them from adverse marine conditions, allow for safe operations of the mobile gates, and provide a good service to all stakeholders operating in Venice. Time will show if the scientific community is able to provide such a system that allows Venice to continue to remain one of the most beautiful and visited cities in the world.

## 7. Conclusions

As shown in the assessment above, there are many operational systems providing storm surge forecasts in Venice. Here are recommendations that could lead to significant improvements in the forecast systems, and that should be implemented or at least tested for Venice:

1.  The basis for a good water level forecast is a reliable meteorological forecast. This is probably the most important factor in improving surge forecasts. Wind fields from ECMWF are probably the best choice that concerns
reliability, but in some cases they are too coarse to resolve small-scale features, both in time and space. Other models (COSMO-I, Bolam, Moloch) provide higher spatial resolution and might resolve these small-scale processes, but assimilation is missing in these models. Satellites can be useful to correct wind fields with scatterometer data.

2.  Ensemble forecasts are extremely useful as they allow for the quantification of uncertainty. Dealing with
uncertainty is of paramount interest in the forecasting business because it allows to alert the population of possible disaster and to make decisions based on this uncertainty in the MOSE operations.

3.  Assimilation of water level measurements needs to be carried forward. Assimilating water levels will allow for better and more precise initial conditions, especially important in the case of forecasting seiches that occur in the



Adriatic Sea. Continuous assimilation of tide gauge data will improve short-range forecasts, especially important for the operations of the MOSE, which needs a precise forecast some hours (4-6 hours) ahead of the water level maximum.

4.   The fact that there are already multiple (independent) models running in Venice should be exploited via a multi-model approach. A multi-model evaluation of the various model outputs would allow a rating of the models and an identification of the most appropriate system for different weather conditions.

Implementing all these recommendations is quite feasible, and this would bring forward the art of storm surge forecasting which would improve control of the MOSE barriers and facilitate better adaptation to floods in the Venice lagoon in general.

**Acknowledgments**

This work was partially supported by the STREAM project (Strategic development of flood management, project ID 10249186) funded by the European Union under the V A Interreg Italy-Croatia CBC programme. Moreover, this scientific
activity was performed with the contribution of the Provveditorato for the Public Works of Veneto, Trentino Alto Adige and Friuli Venezia Giulia, provided through the concessionary of State Consorzio Venezia Nuova and coordinated by CORILA under the project Venezia 2021. The work of M. Orlić has been supported by Croatian Science Foundation under the project IP-2018-01-9849 (MAUD).

**Author contribution**

GU coordinated the paper with help from MB and CF. Specific contributions to the sections are as follows (first author is lead author). Section 1: GU PL RN Section 2.1: AC MB MO Section 2.2: GU Section 2.3: MF EC AP SM Section 3.1: CF Section 3.2: AV Section 3.3: PL AP Section 3.4: MO MB Section 4.1: ABF XB Section 4.2: IDH Section 4.3: JWN Section 4.4: EAF BPG BM Section 4.5: MB AC AV GU CF MO Section 4.6: GU MB CF AT AP MF EC Section 5.1: MF AP EC SM AB Section 5.2: MB AV AC Section 5.3: AT AP Section 5.4: MB XB PL Section 5.5: EAF BPG MB PL Section 5.6: CF AP MF
MB SM EC AB Section 6: GU MB CF PL DZ IDH RN Section 7 GU CF IDH RN

**Competing Interest**

The authors declare that they have no conflict of interest.

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

**Figures and Tables**

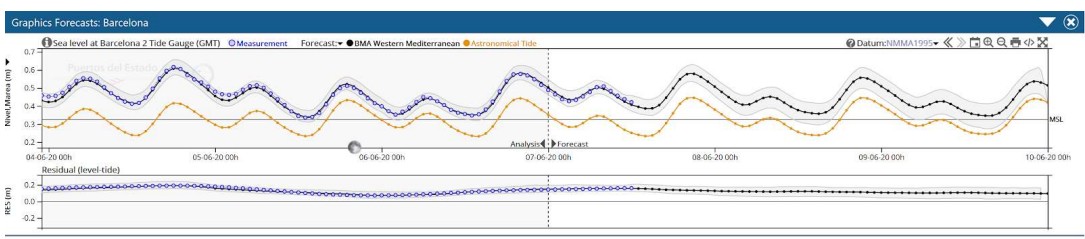


Fig. 1: Sea level forecasted at Barcelona on 7/6/2020 (hourly time series): black: BMA solution (shaded area: 95% uncertainty band based on the output of the different models); blue: tide gauge observations; orange: tide. The BMA solution is obtained by combining the sea level solutions from IBI-MFC, MED-MFC and Nivmar at Barcelona harbour (as displayed in PdE Portus system: https://portus.puertos.es).




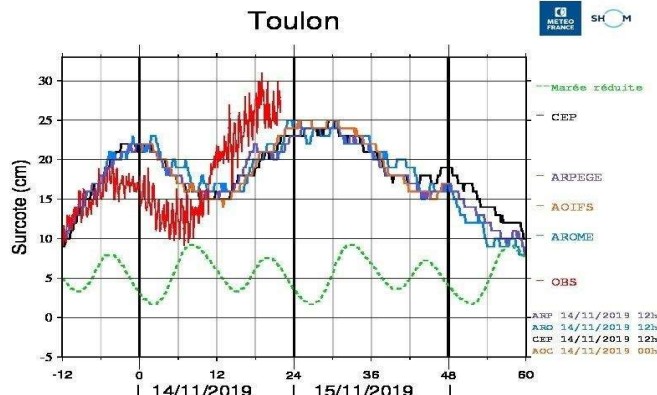

Fig. 2: Temporal series (every 10 minutes) for the storm surge forecasted at Toulon (South of France) starting the 14/11/2019 at 12 UTC, using the forcing provided from IFS (named CEP, black), ARPEGE (purple), AROME (blue) and AROME-IFS (orange). Red: tide gauge observations, green: tide height (normalized).





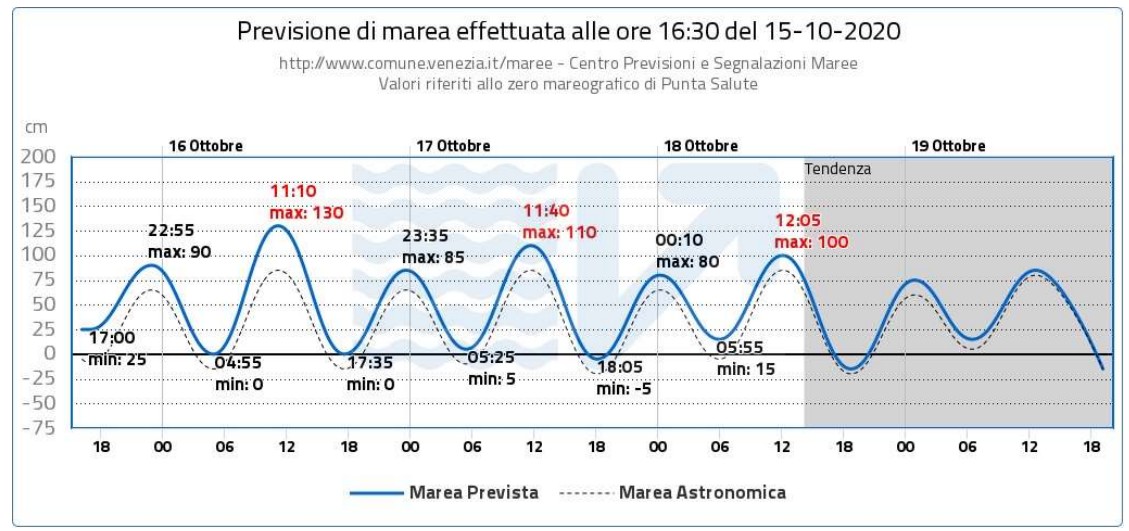

Fig. 3: Operational forecast site of CPSM, the Venice Municipality. An example of the daily bulletin.



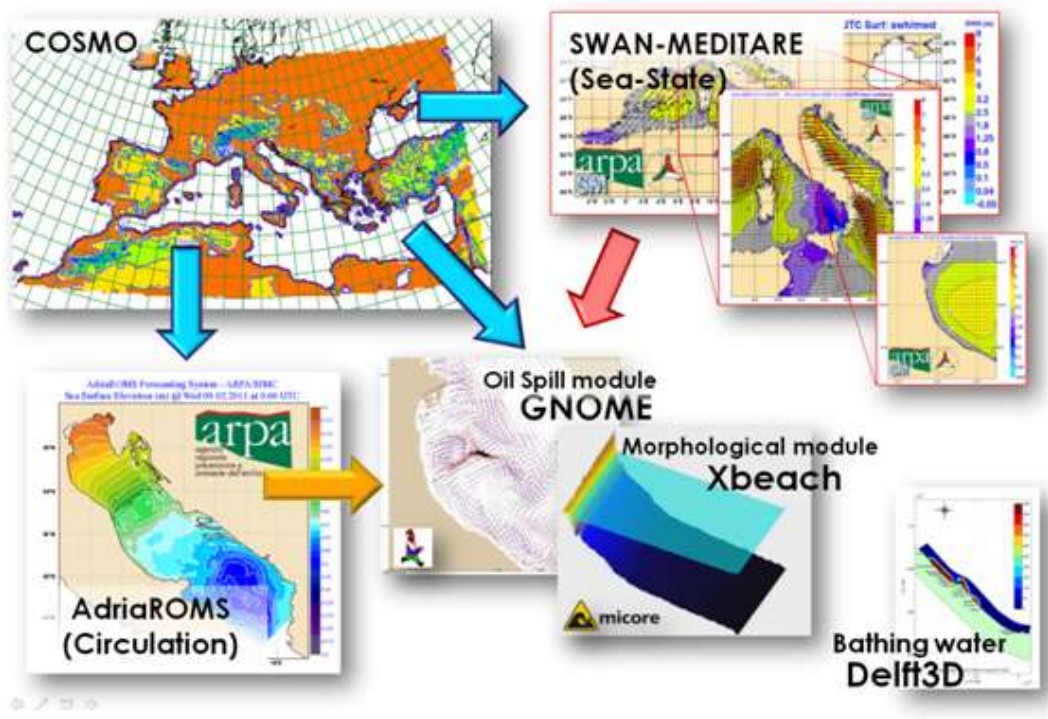


Fig. 4: The operational modelling system and the meteo-marine products provided by Arpae-SIMC.





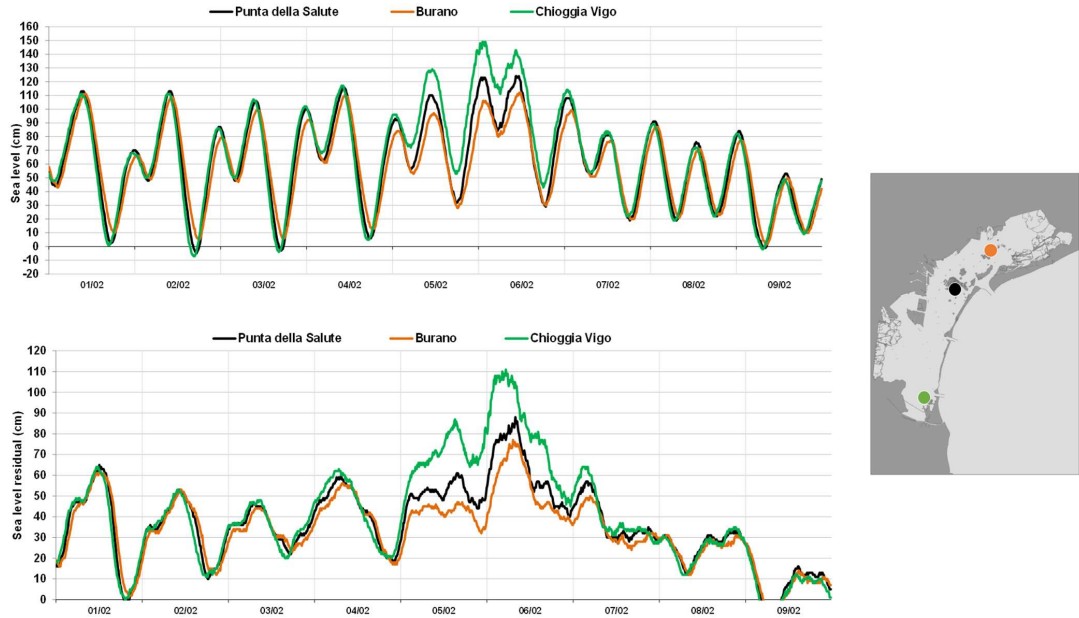

Fig. 5: Sea level (top) and residual level (bottom) at Venice – Punta della Salute (blue), Burano (red), Chioggia (green), during

a Bora event (wind from NE) on February 2015.



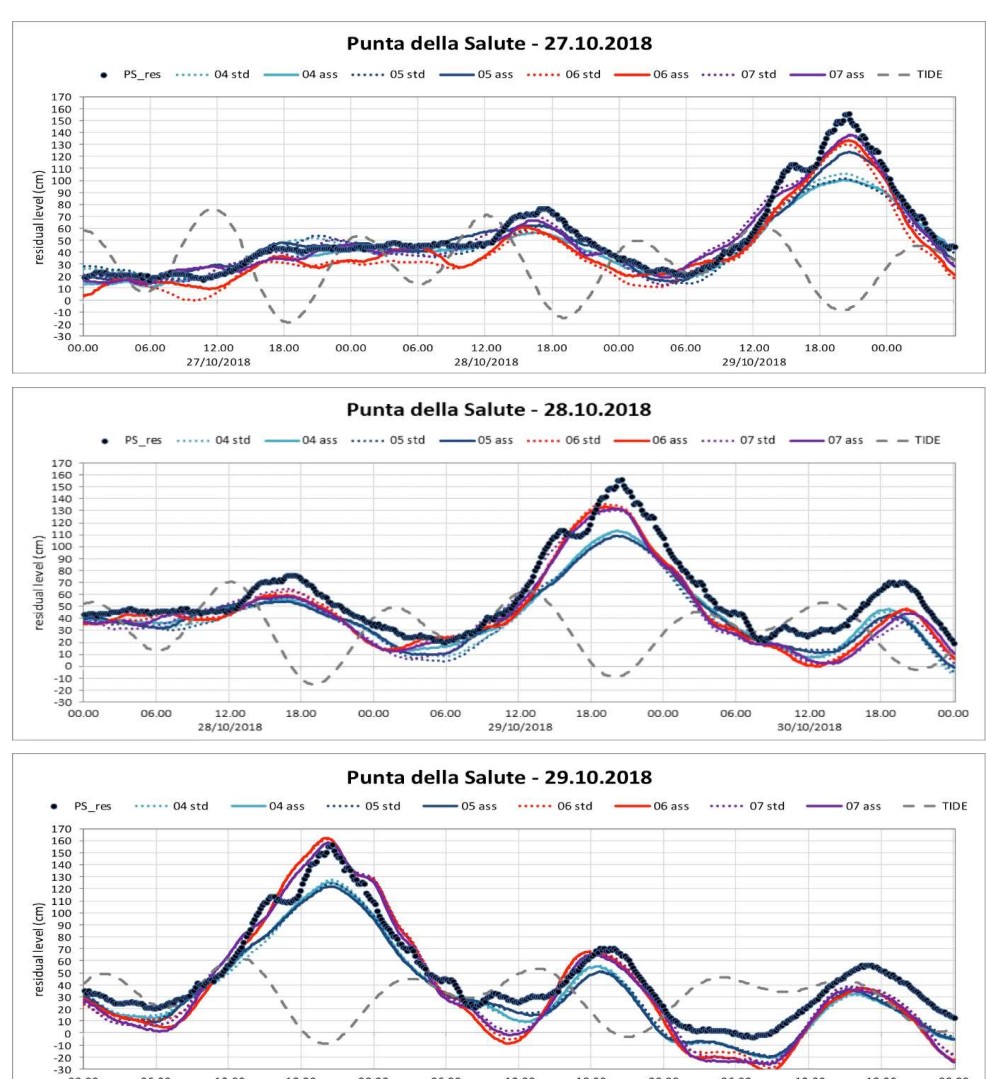

Fig. 6: Forecast for the VAIA storm on the 29th of October, 2018. Results are shown from various versions of the model running at ISPRA. The observed residual is given with black dots. Top panel: forecast 3 days before the event, middle panel: 2 days, bottom panel: 1 day.



| Model Name | Number of coefficient sets | Selection method | Number of regressors | Variables used | Source of meteorological data |
|---|---|---|---|---|---|
| bigcm2 | 141 | (1) | 122 | tide level, atmospheric pressure | Synop data |
| sea | 135 | (1) | 172 | tide level, atmospheric pressure | Synop data |
| mlp | 9 | (1) | 75 | tide level, atmospheric pressure | Synop data |
| sumdb et bigsumdp | 141 | (1) | 75 and 122 | tide level, atmospheric pressure | forecast ECMWF |
| sea4run | 135 | (1) | 122 | tide level, atmospheric pressure | forecast ECMWF |
| mixsum et mixbig | 141 | (1) | 75 and 122 | tide level, atmospheric pressure | forecast ECMWF |
| sumlami et biglami | 141 | (1) | 75 and 122 | tide level, atmospheric pressure | forecast COSMO-LAMI |
| sum et bigwindlami | 165 | (2) | 90 and 110 | tide level, atmospheric pressure, wind | forecast COSMO-LAMI |
| scontraura | 180 | (3) | 122 | tide level, atmospheric pressure, wind | forecast ECMWF |
| scontrayear | 60 | (3) | 75 | tide level, atmospheric pressure, wind | forecast ECMWF |
| Sum, bigensemble et alessamble | 141 | (1) | 75 and 122 | tide level, atmospheric pressure | forecast ECMWF-ensemble |
| Sum et bigtower | 141 | (1) | 75 and 122 | tide level, atmospheric pressure | forecast ECMWF |





Table 1: List of statistical models running at CPSM. The sets of coefficients are selected depending mainly on: (1) maximum pressure gradient between opposite coasts of the Adriatic Sea, seasonality; (2) classes of Sirocco wind along the 4 region of Adriatic Sea, seasonality; (3) classes of Sirocco wind and Bora wind along the multiple regions of Adriatic Sea, seasonality.