# Peer review of "The prediction of floods in Venice: methods, models and uncertainty (review article)"

_Natural Hazards and Earth System Sciences, 2020_

## Referee Comment (RC1)

*The prediction of floods in Venice: methods, models and uncertainty*

Recommendation: Major Revisions.

The manuscript discusses the physics behind the flooding of Venice and the state of the art of the storm surge forecasting in the Adriatic Sea as well as European coasts. Authors draw conclusions and recommendations for storm surge forecasting and its uncertainties in the Adriatic Sea based on several results from other works. Particularly, the focus is made on the improvement of forecast (and its uncertainty) to aid stakeholders in the opening or closing of the building mobile barriers, called "MOSE", that protect Venice.

Although the topic is interesting and they make a deep and adequate description of the oceanographic dynamic of the Adriatic Sea and the state of the art of forecasting, I think the manuscript writing/structure needs further work. The manuscript needs a better organization of the main results related to the aspects of the dynamic and the forecasting systems. This review is critical, nonetheless the authors have the potential to have a great manuscript and I want to encourage them in their progress.

Major comments

1. Due to the aim of the manuscript is to review the Adriatic Sea, Venice lagoon and Venice city dynamic and forecasting, I think a map of the region with the bathymetry and the principal cities is mandatory for people unfamiliar with the area of study. Despite the referring to a figure in another paper, a map will facilitate to understand the general descriptions without the need to visit other works.

2. There are repetitive descriptions of some aspects of the oceanographic dynamic and the atmospheric forcing throughout the manuscript. The authors should avoid the

reiterative descriptions and they should reference to the corresponding section. For instance, wind phenomena are defined at line 99, Section 3.2. and Section 5.1.

3. Even though some values are scattered throughout the manuscript, please provide water level references as tidal range, tidal datums (and its definition) or variability range when you discuss the amplitude of the events. These references will make it easier to understand the impact of the storm surges. For instance at lines: 45, 209, 313.

4. I agree with the need to compare the state of the art of the Adriatic Sea forecasting systems with the European one; however, I don't think it is a good idea to consider a section for an extensive discussion. Section 4 presents a good review of ocean forecasting in Europe but I lost the focus along the reading. Since the Adriatic Sea is the main region, I suggest that Section 4 starts at Subsection 4.5 and its results to be compared with the European forecast systems.

5. As the manuscript reasoning goes, I think Sections 6 and 7 should be together because their discussions are directly linked. In addition, I suggest starting the "Discussion and conclusions" section with the paragraph of line 733. It recaps one the main and current motivations of the storm surge forecasting.

Minor comments

1. Line 34: What does MOSE stand for?
2. Line 42: Acqua Alta should be in italic.
3. Line 42: I assume that there is a typing error in the Bibtex file for Lionello et al.
4. Line 43: Please reword this sentence, I understand the idea but it is twisted.
5. Line 44: Typing error in the Bibtex.
6. Line 57: The comma is not necessary.
7. Line 58: Please rework this sentence, it is hard to understand.

8.  Line 64: It should be "to warn"

9.  Line 64: The comma is not necessary.

10. Line 91: Some reference should be worthwhile.

11. Line 99: Since it is the first time that Sirocco appears, it should be in italic.

12. Line 100: As Sirocco, Bora should be in italic, at least the first time.

13. Line 104: A value of mean or extreme amplitude would be worthwhile.

14. Line 108: … has a peak …

15. Line 108: "with the Adriatic Sea"?

16. Line 113: What is the tidal range?

17. Line 123: The comma is not necessary.

18. Line 125: "However"?

19. Line 125: "the first one" … "the second one"

20. Line 133: What is the tidal range?

21. Line 135: "during" instead "in"

22. Line 140: Please, provide an amplitude value as reference

23. Line 142: Typing error in the Bibtex.

24. Line 143: This paragraph is more adequate for the introduction or the final conclusions.

25. Line 151: Please, provide a cite.

26. Line 169: I assume that it is about Latex typing, homogenize the uses of "".

27. Line 170: Please homogenize the references to atmospheric sea level pressure. I noted many differences throughout the manuscript.

28. Line 173: "For this reason" sounds awkward, maybe "As consequence" or something like that.

29. Line 175: This sentence should be at the beginning of the paragraph.

30. Line 176: Why?

31. Line 189: "reaching gusts of about 100 km/h and up to 200 km/h"

32. Line 181: This paragraph continues with the same idea of the previous one. It should be a single paragraph.

33. Line 184: Space after period.

34. Line 196: Semicolon between references.

35. Line 203: Please, remove the parentheses, both statements should be part of the text.

36. Line 238: What does SRL stand for?

37. Line 257: For unfamiliar readers, it's worthwhile to mention what type of models are FES2012 and GFS. Beside, please clarify what model has a resolution of about 250 m.

38. Line 260: What does "H" stand in "2DH"?

39. Line 264: Please provide a cite for Charnock approach.

40. Line 383: Rissaga should be in italic.

41. Line 424: the phrase inside the parentheses should be as part of the text after the "but".

42. Line 428: Since HYPSE and System based on Delft-3D are no longer operative, it should be as a comment instead part of the list.

43. Line 440: Space between a and Centre.

44. Line 449: "MedFS" should not be in italic.

45. Line 491: Please provide a cite for the Tiresias model.

46. Line 535: Throughout the manuscript storm surge was written without the dash.

47. Line 537: Is there any work about nonlinear interaction between those phenomena in the region?

48. Line 556: Please provide a cite for COSMO-I.

49. Line 558: The period should be after the parentheses.

50. Line 574: Space after the period.

51. Line 610: The definition should be wider. For instance, Flowerdew et al. (2009) say: *Each forecast uses slightly different initial conditions, boundary conditions, and/or model physics (collectively, model inputs), with the aim of sampling the range of forecast results that are consistent with the uncertainty in the model and observations (Palmer, 2006)*.

Flowerdew et al. (2009): https://doi.org/10.1080/01490410902869151

Palmer (2006): "Predictability of weather and climate: from theory to practice". In *Predictability of weather and climate*, Edited by: Palmer, Tim and Hagedorn, Renate. Chapter 1. Cambridge: Cambridge University Press.

52. Line 651: What does ZMPS stand for?

53. Line 681: Typing error in the Bibtex.

54. Line 693: This paragraph should be at the beginning of the subsection.

55. Line 726: Remove the comma after Krzysztofowicz.

---

## Referee Comment (RC2) · Anonymous Referee #2 · 21 Dec 2020

First, I would like to congratulate the authors on their work to providing a comprehensive review of storm surge forecasting methods in the northern Adriatic Sea. The background information on the physics of the flooding behind Venice also provides good foundation for the readers to understand the problem and appreciate the methods that are available. The review is through and covers multiple facets of storm surge forecasting both in a regional and local level.

However, I believe that a few more inputs would improve the manuscript to help provide more information to readers. With that said, I would recommend this paper for publication after minor revisions.

1. It is stated in the manuscript that a reliable meteorological forecast is an important factor. It would be interesting to see your approach on quantifying uncertainty coming

from the forecasts themselves and how big that uncertainty is in relation to others (for instance model uncertainty)

2. In relation to the first question, have you done comparisons between the different forecasts? Or are there a previous studies providing the levels of uncertainties of different forecasts for north Adriatic region?

3. Another interesting input to the paper would be the comparison in performance of the forecast based on numerical models versus forecast based on data-driven models, from former studies.

4. Is a long-term storm surge forecast of interest to this paper? If yes, how useful are the global climate models for that? In terms of the spatial and temporal resolution, and level of uncertainty?

---

## Author Comment (AC1) · 7 Feb 2021

**Review 1**

*Recommendation : Major Revisions.*

*The manuscript discusses the physics behind the flooding of Venice and the state of the art of the storm surge forecasting in the Adriatic Sea as well as European coasts. Authors draw conclusions and recommendations for storm surge forecasting and its uncertainties in the Adriatic Sea based on several results from other works. Particularly, the focus is made on the improvement of forecast (and its uncertainty) to aid stakeholders in the opening or closing of the building mobile barriers, called "MOSE", that protect Venice.*

*Although the topic is interesting and they make a deep and adequate description of the oceanographic dynamic of the Adriatic Sea and the state of the art of forecasting, I think the manuscript writing/structure needs further work. The manuscript needs a better organization of the main results related to the aspects of the dynamic and the forecasting systems. This review is critical, nonetheless the authors have the potential to have a great manuscript and I want to encourage them in their progress.*

We thank the reviewer for his comments and his recommendations. Below we detail how we want to handle the requested changes.

*Major comments*

*1. Due to the aim of the manuscript is to review the Adriatic Sea, Venice lagoon and Venice city dynamic and forecasting, I think a map of the region with the bathymetry and the principal cities is mandatory for people unfamiliar with the area of study. Despite the referring to a figure in another paper, a map will facilitate to understand the general descriptions without the need to visit other works.*

We agree with the reviewer that a map will be very useful, and we will insert one in order to show the physical setting and all the names mentioned in the text.

*2. There are repetitive descriptions of some aspects of the oceanographic dynamic and the atmospheric forcing throughout the manuscript. The authors should avoid the reiterative descriptions and they should reference to the corresponding section. For instance, wind phenomena are defined at line 99, Section 3.2. and Section 5.1.*

Because many authors have contributed to the writing of the manuscript, some repetition arose. We will carefully edit the manuscript and eliminate all places where such repetitions occur.

*3. Even though some values are scattered throughout the manuscript, please provide water level references as tidal range, tidal datums (and its definition) or variability range when you discuss the amplitude of the events. These references will make it easier to understand the impact of the storm surges. For instance at lines: 45, 209, 313.*

We think this is a very good recommendation and we will insert more information about the amplitude of the events. This will make it easier for a reader, not acquainted with the Venice setting, to get a better understand of the importance of single storm surges.

*4. I agree with the need to compare the state of the art of the Adriatic Sea forecasting systems with the European one; however, I don't think it is a good idea to consider a*

*section for an extensive discussion. Section 4 presents a good review of ocean forecasting in Europe but I lost the focus along the reading. Since the Adriatic Sea is the main region, I suggest that Section 4 starts at Subsection 4.5 and its results to be compared with the European forecast systems.*

We understand the worries of the reviewer that the section on European storm surge forecasting systems was too long. However, starting with the discussion of the Adriatic Sea would make the structure of the article still more complicated. After the Adriatic Sea, we would have first to discuss the Venice Lagoon, and only then introduce the European picture. We propose to still start with the European case, but merge section 4.1-4.4 into one first section on European forecasting systems, and having as subsections The Atlantic, North Sea, Baltic Sea and the Mediterranean. This new section will be reduced in length and more streamlined to not disturb the focus on the Adriatic Sea and the Venice Lagoon.

*5. As the manuscript reasoning goes, I think Sections 6 and 7 should be together because their discussions are directly linked. In addition, I suggest starting the "Discussion and conclusions" section with the paragraph of line 733. It recaps one the main and current motivations of the storm surge forecasting.*

As the reviewer has requested, we will merge the discussion and conclusion section into one.

*Minor comments*

We thank the reviewer for the careful reading of the text, and we will handle all requested changes. We are not going to discuss all points here, but only the major ones that have been raised.

*24. Line 143: This paragraph is more adequate for the introduction or the final conclusions.*

We will shift the paragraph as requested.

*37. Line 257: For unfamiliar readers, it's worthwhile to mention what type of models are FES2012 and GFS. Beside, please clarify what model has a resolution of about 250 m.*

Will be done.

*42. Line 428: Since HYPSE and System based on Delft-3D are no longer operative, it should be as a comment instead part of the list.*

Yes, will be done.

*51. Line 610: The definition should be wider. For instance, Flowerdew et al. (2009) say: Each forecast uses slightly different initial conditions, boundary conditions, and/or model physics (collectively, model inputs), with the aim of sampling the range of forecast results that are consistent with the uncertainty in the model and observations (Palmer, 2006).*
*Flowerdew et al. (2009): https://doi.org/10.1080/01490410902869151*
*Palmer (2006): "Predictability of weather and climate: from theory to practice". In Predictability of weather and climate , Edited by: Palmer, Tim and Hagedorn, Renate. Chapter 1. Cambridge: Cambridge University Press.*

Thank you for the information. We will incorporate it into the new revision.

---

## Author Comment (AC2) · 7 Feb 2021

**Review 2**

*First, I would like to congratulate the authors on their work to providing a comprehensive review of storm surge forecasting methods in the northern Adriatic Sea. The background information on the physics of the flooding behind Venice also provides good foundation for the readers to understand the problem and appreciate the methods that are available. The review is through and covers multiple facets of storm surge forecasting both in a regional and local level.*
*However, I believe that a few more inputs would improve the manuscript to help provide more information to readers. With that said, I would recommend this paper for publication after minor revisions.*

We thank the reviewer for the positive evaluation of our manuscript and for the precious recommendations. Below we detail how we are going to integrate these comments.

*1. It is stated in the manuscript that a reliable meteorological forecast is an important factor. It would be interesting to see your approach on quantifying uncertainty coming from the forecasts themselves and how big that uncertainty is in relation to others (for instance model uncertainty)*

This important question has to be resolved by the meteorological community. We will search for existing information that would indicate the inherent uncertainty of the meteorological data. We will discuss this point also in the discussion section where the review can make recommendations for further research and action.

*2. In relation to the first question, have you done comparisons between the different forecasts? Or are there a previous studies providing the levels of uncertainties of different forecasts for north Adriatic region?*

This is one of the important points that a forecasting system has to provide. The only published data on this is the i-Storms initiative that uses multi-model ensemble forecasting that allows an assessment on uncertainty. We will stress this point more thoroughly in the discussion.

*3. Another interesting input to the paper would be the comparison in performance of the forecast based on numerical models versus forecast based on data-driven models, from former studies.*

Some papers exist that provide an analysis of the score of various forecast systems. We will insert these findings into the manuscript and discuss them there.

*4. Is a long-term storm surge forecast of interest to this paper? If yes, how useful are the global climate models for that? In terms of the spatial and temporal resolution, and level of uncertainty?*

The implication of the impact of climate change on the frequency and strength of storm surges on the Venice lagoon has already been discussed in other papers published by authors of this manuscript. We will insert some of these findings into the discussion section of the manuscript.

---

## Author Response (AR1)

Dear Editor,

We are glad to submit a new version of our manuscript. We have addressed all the comments of the reviewers. Please find below a detailed response to all the points that have been raised. We have rewritten and condensed sections 4.1-4.4, and have also mainly rewritten the discussion and conclusion section. Moreover, we added two new figures to show the setting of Venice and the Adriatic Sea and better explain the water level datum used in Venice, together with storm surge levels of past storms.

I hope our manuscript will now be acceptable for publication in NHESS.

Sincerely, Georg Umgiesser (on behalf of all co-authors)

**Review 1**

*Recommendation: Major Revisions.*

*The manuscript discusses the physics behind the flooding of Venice and the state of the art of the storm surge forecasting in the Adriatic Sea as well as European coasts. Authors draw conclusions and recommendations for storm surge forecasting and its uncertainties in the Adriatic Sea based on several results from other works. Particularly, the focus is made on the improvement of forecast (and its uncertainty) to aid stakeholders in the opening or closing of the building mobile barriers, called "MOSE", that protect Venice.*

*Although the topic is interesting and they make a deep and adequate description of the oceanographic dynamic of the Adriatic Sea and the state of the art of forecasting, I think the manuscript writing/structure needs further work. The manuscript needs a better organization of the main results related to the aspects of the dynamic and the forecasting systems. This review is critical, nonetheless the authors have the potential to have a great manuscript and I want to encourage them in their progress.*

We thank the reviewer for his comments and his recommendations. Below we detail how we handled the requested changes.

*Major comments*

*1. Due to the aim of the manuscript is to review the Adriatic Sea, Venice lagoon and Venice city dynamic and forecasting, I think a map of the region with the bathymetry and the principal cities is mandatory for people unfamiliar with the area of study. Despite the referring to a figure in another paper, a map will facilitate to understand the general descriptions without the need to visit other works.*

We agree with the reviewer that a map will be very useful. We therefore have inserted a new figure (Fig.1) that shows the setting of the Venice Lagoon and the Adriatic Sea.

*2. There are repetitive descriptions of some aspects of the oceanographic dynamic and the atmospheric forcing throughout the manuscript. The authors should avoid the reiterative descriptions and they should reference to the corresponding section. For instance, wind phenomena are defined at line 99, Section 3.2. and Section 5.1.*

Because many authors have contributed to the writing of the manuscript, some repetition arose. We have tried to eliminate most of the repetitions, which mainly occurred when talking about the Adriatic Sea, the tides and the winds. However, some small overlaps have remained, and, to our opinion, cannot be eliminated.

*3. Even though some values are scattered throughout the manuscript, please provide water level references as tidal range, tidal datums (and its definition) or variability range when you discuss the amplitude of the events. These references will make it easier to understand the impact of the storm surges. For instance at lines: 45, 209, 313.*

We think this is a very good recommendation. We have therefore added a new figure (Fig. 2) that shows exactly the various water levels and the datum for the city of Venice. We have also inserted references to this figure in various parts of the text and additionally discussed the issue, such as section 1, 3.1, 3.4, 5.6

*4. I agree with the need to compare the state of the art of the Adriatic Sea forecasting systems with the European one; however, I don't think it is a good idea to consider a section for an extensive discussion. Section 4 presents a good review of ocean forecasting in Europe but I lost the focus along the reading. Since the Adriatic Sea is the main region, I suggest that Section 4 starts at Subsection 4.5 and its results to be compared with the European forecast systems.*

We understand the worries of the reviewer that the section on European storm surge forecasting systems was too long. However, starting with the discussion of the Adriatic Sea would make the structure of the article still more complicated. After the Adriatic Sea, we would have first to discuss the Venice Lagoon, and only then introduce the European picture. We have therefore merged sections 4.1-4.4 into one first section 4.1 on European forecasting systems, having subsections 4.1.1-4.1.4 as The Atlantic, North Sea, Baltic Sea and the Mediterranean. We have rewritten, condensed, and streamlined the original 140 lines to 100 lines in the revised text. In this way, it will not disturb the focus on the Adriatic Sea and the Venice Lagoon. I hope this is acceptable for the reviewer.

*5. As the manuscript reasoning goes, I think Sections 6 and 7 should be together because their discussions are directly linked. In addition, I suggest starting the "Discussion and conclusions" section with the paragraph of line 733. It recaps one the main and current motivations of the storm surge forecasting.*

As the reviewer has requested, we have merged the discussion and conclusion section into one final section called "Discussion".

*Minor comments*

We thank the reviewer for the careful reading of the text, and we have handled all requested changes. We have detailed our changes below.

1. Line 34: What does MOSE stand for?

MOdulo Sperimentale Elettromeccanico, Experimental Electromechanical Module. We have inserted this explicatory text.

2. Line 42: Acqua Alta should be in italic.

Ok, changed.

3. Line 42: I assume that there is a typing error in the Bibtex file for Lionello et al.

This paper is part of a special issue on Venice, and the other papers are cited like this. I will have to consult with the editor if we can change the reference to something different.

4. Line 43: Please reword this sentence, I understand the idea but it is twisted.

Changed.

5. Line 44: Typing error in the Bibtex.

Please see response to comment 3.

6. Line 57: The comma is not necessary.

Ok

7. Line 58: Please rework this sentence, it is hard to understand.

We have reformulated the sentence to make it clearer.

8. Line 64: It should be "to warn"

Ok

9. Line 64: The comma is not necessary.

Ok

10. Line 91: Some reference should be worthwhile.

We have inserted a reference and changed the sentence to:

The order of magnitude of the surface boundary currents (about 20 cm/s, Poulain, 2001) suggests that changes of sea surface elevation induced by geostrophic balance are negligible.

11. Line 99: Since it is the first time that Sirocco appears, it should be in italic.
12. Line 100: As Sirocco, Bora should be in italic, at least the first time.

We have put the first mention of Sirocco and Bora in italics.

13. Line 104: A value of mean or extreme amplitude would be worthwhile.

We have given the numbers (50 cm)

14. Line 108: … has a peak …
15. Line 108: "with the Adriatic Sea"?

Corrected and reformulated.

16. Line 113: What is the tidal range?

Added (1 m)

17. Line 123: The comma is not necessary.

Ok

18. Line 125: "However"?

Deleted.

19. Line 125: "the first one" … "the second one"

Ok, changed.

20. Line 133: What is the tidal range?

We added this information some lines above, so I would not repeat this information.

21. Line 135: "during" instead "in"

Changed.

22. Line 140: Please, provide an amplitude value as reference

We have added a reference to Fig 2, where all the information is contained.

23. Line 142: Typing error in the Bibtex.

See response to comment 3.

24. Line 143: This paragraph is more adequate for the introduction or the final conclusions.

This paragraph, as requested, has been shifted to the Introduction section.

25. Line 151: Please, provide a cite.

We have inserted a reference (Tsimplis et al., 1995) for this statement.

26. Line 169: I assume that it is about Latex typing, homogenize the uses of "".

We have deleted the use of "" when referring to winds.

27. Line 170: Please homogenize the references to atmospheric sea level pressure. I noted many differences throughout the manuscript.

We have substituted the term sea level pressure by atmospheric pressure in various parts of the manuscript.

28. Line 173: "For this reason" sounds awkward, maybe "As consequence" or something like that.

We have changed the text to "As a consequence"

29. Line 175: This sentence should be at the beginning of the paragraph.

We have shifted it to the beginning of the paragraph.

30. Line 176: Why?

It is a matter of fact. We have slightly reformulated the sentence.

31. Line 179: "reaching gusts of about 100 km/h and up to 200 km/h"

Corrected.

32. Line 181: This paragraph continues with the same idea of the previous one. It should be a single paragraph.

Since in this second paragraph we describe the two different types of Bora winds, we would really like to keep them separate.

33. Line 184: Space after period.

Ok

34. Line 196: Semicolon between references.

Ok

35. Line 203: Please, remove the parentheses, both statements should be part of the text.

Parenthesis removed.

36. Line 238: What does SRL stand for?

This was the only reference to SLR, so we spelled out the acronym: sea level rise.

37. Line 257: For unfamiliar readers, it's worthwhile to mention what type of models are FES2012 and GFS. Beside, please clarify what model has a resolution of about 250 m

We have clarified the acronyms and have added some more references.

38. Line 260: What does "H" stand in "2DH"?

Changed to 2D. The whole paragraph has, however, been rewritten.

39. Line 264: Please provide a cite for Charnock approach.

We have rephrased the sentence and added a reference (Charnock, 1955).

40. Line 383: Rissaga should be in italic.

This paragraph has been removed in order to condense the information on the European forecasting systems.

41. Line 424: the phrase inside the parentheses should be as part of the text after the "but".

Done.

42. Line 428: Since HYPSE and System based on Delft-3D are no longer operative, it should be as a comment instead part of the list.

We have re-written the paragraph and only mention the other two systems for historical reasons.

43. Line 440: Space between a and Centre.

Corrected

44. Line 449: "MedFS" should not be in italic.

Ok

45. Line 491: Please provide a cite for the Tiresias model.

The citation was at the end of the paragraph. We have shifted it up.

46. Line 535: Throughout the manuscript storm surge was written without the dash.

Corrected.

47. Line 537: Is there any work about nonlinear interaction between those phenomena in the region?

We have refered to the paper of Lionello et al., 2021 in the same issue.

48. Line 556: Please provide a cite for COSMO-I.

We have added the reference Schaettler et al. (2018) as a reference for COSMO-I

49. Line 558: The period should be after the parentheses.

Ok

50. Line 574: Space after the period.

Ok

51. Line 610: The definition should be wider. For instance, Flowerdew et al. (2009) say:
*Each forecast uses slightly different initial conditions, boundary conditions, and/or model physics (collectively, model inputs), with the aim of sampling the range of forecast results that are consistent with the uncertainty in the model and observations (Palmer, 2006).*
Flowerdew et al. (2009): https://doi.org/10.1080/01490410902869151
Palmer (2006): "Predictability of weather and climate: from theory to practice". In *Predictability of weather and climate*, Edited by: Palmer, Tim and Hagedorn, Renate. Chapter 1. Cambridge: Cambridge University Press.

The whole section 5.5 on uncertainty has been rewritten, and this comment and its references have been taken into account.

52. Line 651: What does ZMPS stand for?

It was zero mareografico Punta Salute. We have now substituted it with "over datum" and have cited Fig. 2.

53. Line 681: Typing error in the Bibtex.

We have substituted the reference to Ferrarin et al., 2021

54. Line 693: This paragraph should be at the beginning of the subsection.

We have shifted this paragraph into the Discussion section, where it is more appropriate.

55. Line 726: Remove the comma after Krzysztofowicz.

done

**Review 2**

Please find below our responses to the comments of the reviwer.

*First, I would like to congratulate the authors on their work to providing a comprehensive review of storm surge forecasting methods in the northern Adriatic Sea. The background information on the physics of the flooding behind Venice also provides good foundation for the readers to understand the problem and appreciate the methods that are available. The review is through and covers multiple facets of storm surge forecasting both in a regional and local level.*
*However, I believe that a few more inputs would improve the manuscript to help provide more information to readers. With that said, I would recommend this paper for publication after minor revisions.*

We thank the reviewer for the positive evaluation of our manuscript and for the precious recommendations. Below we detail how we addressed the critiques raised.

*1. It is stated in the manuscript that a reliable meteorological forecast is an important factor. It would be interesting to see your approach on quantifying uncertainty coming from the forecasts themselves and how big that uncertainty is in relation to others (for instance model uncertainty)*

This important question has to be resolved by the meteorological community. We have inserted a new paragraph that addresses this issue at the beginning of section 4 on storm surge modeling.

*2. In relation to the first question, have you done comparisons between the different forecasts? Or are there a previous studies providing the levels of uncertainties of different forecasts for north Adriatic region?*

This is one of the important points that a forecasting system has to provide. The only published data on this is the i-Storms initiative that uses multi-model ensemble forecasting

that allows an assessment on uncertainty. We have inserted a new paragraph in the discussion section on this point.

*3. Another interesting input to the paper would be the comparison in performance of the forecast based on numerical models versus forecast based on data-driven models, from former studies.*

To our knowledge there is only one paper discussing this issue (Zampato et al., 2016) We have inserted some more discussion on this point where we also discuss .the multi-model approach of comment 2 above. Comparisons with data-driven models have not been published.

*4. Is a long-term storm surge forecast of interest to this paper? If yes, how useful are the global climate models for that? In terms of the spatial and temporal resolution, and level of uncertainty?*

The implication of the impact of climate change on the frequency and strength of storm surges on the Venice lagoon is being discussed in another paper in the same issue of this journal (Lionello et al.). Cross-references are also provided in the manuscript. We therefore do not see the need to discuss this point here in this paper.